# Adversarial Attacks Leverage Interference Between Features in Superposition

## Abstract

Fundamental questions remain about when and why adversarial examples (AExs) arise in neural networks (NNs), with competing views characterising them either as artifacts of the irregularities in the decision landscape or as products of sensitivity to non-robust input features. In this paper, we instead argue that adversarial vulnerability can stem from *efficient* information encoding in NNs. Specifically, we show how superposition – where networks represent more features than they have dimensions – creates arrangements of latent representations that adversaries can exploit. We demonstrate that adversarial perturbations leverage interference between superposed features, making attack patterns predictable from feature arrangements. Our framework provides a mechanistic explanation for two known phenomena: adversarial attack transferability between models with similar training regimes and class-specific vulnerability patterns. In synthetic settings with precisely controlled superposition, we establish that superposition *suffices* to create adversarial vulnerability. We then demonstrate that these findings persist in a vision transformer (ViT) trained on CIFAR-10. These findings reveal adversarial vulnerability can be a byproduct of networks' representational compression, rather than flaws in the learning process or non-robust inputs.

## 1 Introduction

Despite extensive research on AExs (Szegedy et al., 2014; Goodfellow et al., 2014), there remains no consensus on their fundamental cause. We still lack a complete explanation for how minimal perturbations so drastically alter model predictions, leaving us unable to predict which perturbations succeed, why attacks transfer between models, or how to design effective defences.

Existing explanations broadly fit into two camps (Nakkiran, 2019). The *bug* perspective attributes vulnerability to factors including learning flaws (Schmidt et al., 2018; Nakkiran, 2019) or geometric properties of high-dimensional decision boundaries (Fawzi et al., 2016). Work in this domain focuses on providing statistical bounds on properties such as capacity (Bubeck & Sellke, 2021) or Lipschitz constraints (Hein & Andriushchenko, 2017) without explaining which specific perturbations succeed. The *feature* perspective argues AExs exploit predictive but non-robust statistical patterns in data (Ilyas et al., 2019), reframing the debate to how underlying data relates to human perception. It treats these non-robust features as fixed properties of the data rather than arising from network encoding strategies. Neither approach reconciles how representational constraints interact with data semantics.

In this paper, we bridge this divide through a mechanistic interpretability perspective, demonstrating how adversarial vulnerability emerges from the interaction between architectural constraints and data semantics (*i.e.* human interpretable properties of the data). We show that AExs can emerge from interference between learned representations in NNs. Adversarial attacks systematically exploit superposed latent structure – a mechanism that normally enables additional representation capacity – to craft effective perturbations that manipulate model outputs. Our analysis reveals a mechanistic pathway: input correlations constrain feature arrangements, these arrangements determine interference patterns, and these interference patterns dictate attack characteristics and transferability. This framework enables prediction of which perturbations succeed and why they transfer between models.

Our account of AExs draws on recent theories about how networks encode and process information, namely the linear representation hypothesis (LRH) (Park et al., 2024) and the theory of superposition (Elhage et al., 2022). The LRH posits that *input features*—fundamental abstractions of data—are

represented as linear directions in a network's representation space. It is hypothesised that NNs can represent significantly more of these features than they have neurons through superposition, enabling networks to efficiently pack multiple features into shared dimensions at the cost of introducing interference. Such interference means perturbing one feature can affect others in non-obvious ways. This paper investigates whether this interference creates vulnerabilities that AExs exploit, and what insights this offers for understanding adversarial phenomena. Our primary contributions are:

- We present a framework reconciling architectural constraints with feature properties in adversarial vulnerability. Using simple models, we demonstrate how data properties induce specific superposition geometries that directly determine realised adversarial perturbations. We show that interference patterns offer insights into attack transferability and class-specific susceptibility.
- We demonstrate these findings persist in a ViT trained on CIFAR-10 with an engineered bottleneck layer to induce superposition.
- We show superposition to be sufficient, but not necessary, for adversarial vulnerability. In identifying this, we establish algorithmic brittleness as a distinct vulnerability mechanism.
- We demonstrate that mechanistic understanding of learned representations enables informed adversarial attack construction, underpinning our argument for semantically-informed defences.

## 2 BACKGROUND

We now provide an overview of the tools we use in our analysis: the LRH and superposition. Formally, let $\mathbf{x} \in \mathcal{X}$ denote the input to a NN and $\mathbf{h}^{(l)} \in \mathbb{R}^{d_l}$ the activation vector of the $l$-th layer.

**Linear representation hypothesis (LRH)**. The LRH posits that NNs represent many variables of their computation, such as semantic properties of their inputs, as linear directions in their activation space, which can be used as abstractions for reasoning (Park et al., 2024; Guerner et al., 2023). This bias towards representing linear features is hypothesised because linear separability allows networks to easily recognise and manipulate features, and because dot products with subsequent layer weights efficiently process such directional features. Growing research supports this (Gurnee et al., 2023; Park et al., 2024). Let $\mathcal{C} = \{c_1, \ldots, c_M\}$ denote a set of $M$ semantically meaningful latent features (*e.g.* concepts like "presence of shape," or "indoor vs. outdoor"). Formally:

**Definition 1 (Linear Representation Hypothesis (LRH))** *A neural network layer with activations* $\mathbf{h}^{(l)} \in \mathbb{R}^{d_l}$ *satisfies the LRH if it represents the latent features* $\mathcal{C} = \{c_1, \ldots, c_M\}$ *as linear directions* $\{\mathbf{v}_j\}_{j=1}^M \subset \mathbb{R}^{d_l}$ *such that:*

$$\mathbf{h}^{(l)}(\mathbf{x}) \approx \sum_{j=1}^M a_j(\mathbf{x})\, \mathbf{v}_j$$

*where* $a_j(\mathbf{x}) \in \mathbb{R}$ *represents the activation magnitude of feature* $c_j$, *and* $\mathbf{v}_j$ *is the corresponding linear direction. The features are **linearly accessible** (Costa et al., 2025): inputs* $\mathbf{x}_0, \mathbf{x}_1 \in \mathcal{X}$ *that differ mainly in the value of feature* $c_j$ *while holding other features approximately constant (i.e.,* $a_j(\mathbf{x}_1) - a_j(\mathbf{x}_0) = k$ *and* $|a_i(\mathbf{x}_1) - a_i(\mathbf{x}_0)| < \lambda$ *for all* $i \neq j$ *and some small* $\lambda > 0$*) satisfy:*

$$\mathbf{h}^{(l)}(\mathbf{x}_1) - \mathbf{h}^{(l)}(\mathbf{x}_0) \approx k\mathbf{v}_j$$

*where* $k$ *reflects the change in* $c_j$.

**Superposition and sparsity**. NNs are capable of representing many more latent features than there are available dimensions in activation space: $M > d_l$. For example, LLMs can reference many more place names than they have residual stream dimensions. One framework for analysing this phenomenon is superposition, according to which networks use an overcomplete and non-orthogonal set of feature directions $\{\mathbf{v}_j\}_{j=1}^M$. This leverages the fact that $2^{\Theta(d\epsilon^2)}$ almost orthogonal vectors ($< \epsilon$ cosine similarity) can be represented in $d$-dimensional space (Tkocz, 2012), and that sparse vectors can be accurately recovered after projection into lower-dimensional spaces (Elhage et al., 2022; Bereska & Gavves, 2024; Sawmya et al., 2025). This creates two challenges. First, *polysemanticity* Scherlis et al. (2022); Lecomte et al. (2023); Arora et al. (2018) emerges where individual neurons contribute to multiple different features, meaning a neuron's activation does not correspond to a single interpretable concept. Second, non-orthogonal feature directions create *interference* between features – activating one feature activates others. Networks can mitigate these issues through non-linear operations (*e.g.* ReLU, softmax) that disambiguate superposed features (Gurnee et al., 2023).

**Definition 2 (Superposition Hypothesis)** *A network layer represents features in superposition if:*

1. *The number of latent features exceeds layer dimensionality:* $M > d_l$
2. *There exists non-orthogonal feature directions:* $\exists\, i, j$ *with* $i \neq j$ *such that* $\mathbf{v}_i \cdot \mathbf{v}_j \neq 0$
3. *Latent feature activations are sparse:* $\mathbb{E}_{\mathbf{x}}[\|\mathbf{a}(\mathbf{x})\|_0] \ll M$, *where* $\mathbf{a}(\mathbf{x}) = [a_1(\mathbf{x}), \ldots, a_M(\mathbf{x})]^T$

*The network trades representational capacity against feature interference by packing more features than dimensions.*

**Adversarial attacks**. Next we define the notion of adversarial perturbations and attacks following (Szegedy et al., 2014; Costa et al., 2024):

**Definition 3 (Adversarial attack)** *Let* $\mathbb{B}_p(\mathbf{x}, \epsilon)$ *denote the* $\epsilon$-*ball around* $\mathbf{x} \in \mathcal{X}$ *in the* $\ell_p$-*sense, e.g.*, $\mathbb{B}_\infty$ *is the* $\ell_\infty$-*ball. Any small* $\boldsymbol{\delta}$ *that* $p$-*perturbs* $\mathbf{x}$ *to create an* ***adversarial input*** $\mathbf{x}' \in \mathbb{B}_p(\mathbf{x}, \epsilon)$ *such that* $f(\mathbf{x}') \neq f(\mathbf{x})$ *(untargeted) or* $f(\mathbf{x}') = y_{\text{target}}$ *(targeted) is known to be an* ***adversarial attack***. $\ell_\infty$- *or* $\ell_2$-*norm attacks are created via the projected gradient descent (PGD) (detailed in App. A).*

**Definition 4 (Adversarial robustness)** *A classifier* $g_w(\cdot)$ *with parameters* $w$ *is said to be* ***robust*** *at* $\mathbf{x}$ *with radius* $\epsilon > 0$, *if* $g_w(\mathbf{x}') = g_w(\mathbf{x})\ \forall \mathbf{x}' \in \mathbb{B}(\mathbf{x}, \epsilon)$.

## 3  SUPERPOSITION GEOMETRY CAN DETERMINE ADVERSARIAL ATTACKS

We begin by investigating if and how adversarial attacks exploit the interference rooted in the superposition of latent features. Specifically, we ask three questions:

> **1**. *Do adversarial perturbations exploit the interference between superposed features?*
> **2**. *Do correlations in the input shape the geometric arrangement of superposed latent features?*
> **3**. *Can shared latent geometry explain why attacks transfer between independently trained models?*

**Experimental setup**. To investigate these questions, we design a synthetic task that exemplifies established LRH and superposition phenomena Park et al. (2024); Elhage et al. (2022) in a precisely controllable setting. The task is designed to: (1) provide an intuitive classification setting for studying AExs; (2) represent class concepts as linear directions per the LRH; (3) induce controlled superposition between these latent features; and (4) retain a priori knowledge of how inputs correspond to the superposed features — enabling testable predictions about adversarial mechanisms.

Specifically, we partition input $\mathbf{x} \in \mathbb{R}^d$ into $k$ equally-sized groups $\mathbf{x} = [\mathbf{x}^{(1)}, \ldots, \mathbf{x}^{(k)}]$, where $\mathbf{x}^{(j)}$ represents the $p$ input features associated with class $j$. The task identifies which group has the largest sum: $y = \operatorname{argmax}_{j \in \{1, \ldots, k\}} \sum_{i=1}^{p} x_i^{(j)}$. Intuitively, each $x_i^{(j)}$ represents evidence for class $j$, with the $p$ features capturing different class attributes. Following Elhage et al. (2022), we sample $x_i^{(j)} \sim \text{Uniform}(0, 1)$ with sparsity $S$ (probability of setting to zero).

A two-layer network compresses the input into $m$ dimensions through a bottleneck encoder, $\mathbf{h} = \text{ReLU}(\mathbf{W}_e \mathbf{x} + \mathbf{b}_e) \in \mathbb{R}^m$ (where $m < k < d$), and reconstructs them via a decoder, $\mathbf{z} = \mathbf{W}_d \mathbf{h} + \mathbf{b}_d \in \mathbb{R}^k$. Our primary setup uses cross entropy (CE) loss without ReLUs or biases. This is chosen for its simplicity, with the softmax limiting interference by exponentially suppressing contributions from interfering latent features. To ensure our findings are not specific to this context, we validate our findings using a second setup with mean squared error (MSE) loss with ReLUs and biases. The representational challenge is to compress information from $d = k \times p$ dimensions to $m < k$ dimensions, and then recover sufficient information to identify the class with the largest sum.

While non-synthetic tasks will have features representing nuanced concepts that require discovery and interpretation, this setup provides us with ground truth class labels for the latent features. This allows us to compute the true class after any perturbation by recalculating group sums, facilitating our definition of an AEx as satisfying the following two conditions:

1. The model's prediction under perturbation changes (*e.g.* from class A to B)
2. The true class remains unchanged (*e.g.* still class A)

This distinguishes genuine AExs from perturbations that actually change the ground truth (*i.e.* change which group has the highest sum). Attacks are created using $\ell_\infty$- and $\ell_2$-norm PGD.

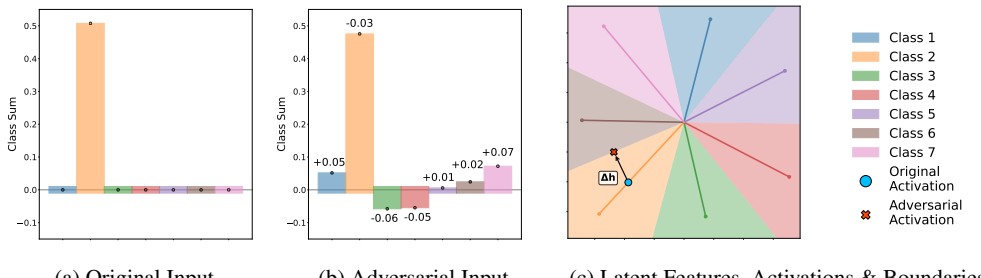

(a) Original Input   (b) Adversarial Input   (c) Latent Features, Activations & Boundaries

Figure 1: **An adversarial attack exploiting superposition geometry** ($k = 7$, $m = 2$). **(a)** The original sample. **(b)** The adversarially perturbed sample, whose ground truth remains the same but is misclassified. The sign and magnitude of an input perturbation is determined by the configuration of latent representations. **(c)** The original and adversarial sample in activation space. The arrows are the column vectors of $\mathbf{W}_e$, the latent representations of the input features.

**Evaluation metrics.** We interpret the columns of the encoder weight matrix $\mathbf{W}_e$ as the latent representations $\mathbf{v}_i$ for each input feature $x_i$. Since input feature $x_i^{(j)}$ contributes exclusively to class $j$, these $\mathbf{v}_i$ (for $i \in \text{class } j$) align. The direct correspondence between input features and class-aligned latent directions allows us to precisely trace how perturbations to the inputs of a specific class, $\mathbf{x}^{(j)}$, relate to changes in latent space. For successful PGD-generated AExs $\mathbf{x}_{adv} = \mathbf{x}_{orig} + \boldsymbol{\delta}$, we measure:

1. **Input perturbation profile (IPP):** The sign and magnitude of each $\delta_i$, describing how different input features are perturbed in an attack.
2. **Latent attack alignment:** The similarity $\Delta \mathbf{h} \cdot \mathbf{v}_c$ between the *latent attack vector* $\Delta \mathbf{h} = \mathbf{h}_{adv} - \mathbf{h}_{orig}$ and class $c$'s representation.
3. **Attack transferability:** Success rate of attacks generated on one model when applied to another.
4. **Robust accuracy:** The fraction of examples that remain correctly classified after perturbation $\delta$ with $\|\delta\|_p \le \epsilon$ to a sample of inputs.

## 3.1 RESULTS

**Do attacks exploit interference between latent features in superposition?** An attack must move a sample across a decision boundary to change its class, but what determines the required input perturbations? Two contrasting intuitions exist: the *feature intuition* suggests increasing input features of the target class, while the *bug intuition* suggests adversarial perturbations modify inputs in unpredictable ways, with knowledge of feature representations providing no insight.

Fig. 1 shows a typical AEx in our setting, showing an input of Class 2 (orange) and its perturbed value (with IPP) that misclassifies it as Class 6 (brown). The perturbations appear arbitrary – we clearly do not simply increase the target class features, contradicting the feature intuition. Yet these perturbations are not random either. Instead, they follow a precise pattern with a systematic correspondence between $\boldsymbol{\delta}$ and the configuration of latent features. This relationship suggests a third intuition: *attacks are mediated by latent geometry*.

Quantitatively, the IPP displays a strong correlation with the latent attack alignment. The sign and magnitude of perturbations $\delta^{(j)}$ correlate strongly with how their class's latent representation $\mathbf{v}^{(j)}$ aligns with the latent attack vector $\Delta \mathbf{h}$. Positively aligned representations ($\mathbf{v}^{(j)} \cdot \Delta \mathbf{h} > 0$) see proportionally amplified input features, while negatively aligned representations experience attenuation. This is what Fig. 1(b) illustrates – input perturbations determined by the dot products in Fig. 1(c) to move the sample across the decision boundary.

Table 1: Alignment between PGD-discovered attacks and theoretically optimal perturbations across configurations. Results show mean $\pm$ std over 1000 attacks per condition. All $p$-values are below $10^{-10}$.

| $k$ | $m$ | Cosine Sim. (PGD vs Theory) | Cosine Sim. (Random Baseline) |
|---|---|---|---|
| 6 | 2 | $0.97 \pm 0.02$ | $0.00 \pm 0.02$ |
| 30 | 10 | $0.96 \pm 0.00$ | $0.00 \pm 0.01$ |
| 90 | 30 | $0.92 \pm 0.00$ | $0.00 \pm 0.00$ |

To evaluate whether PGD attacks specifically exploit superposition geometry, we compare PGD-generated attacks against theoretically optimal perturbations that we show leverage superposition

(derived in Section Sec. 3.2). For each configuration $(k, m)$, we train 5 models with different random seeds and generate 1000 PGD attacks per model, calculating the cosine similarity between each successful perturbation and the corresponding optimum. We establish a random baseline by generating perturbations with matching $\ell_2$ norm and computing their similarity to the theoretical optimum. Tab. 1 shows near-perfect alignment between PGD and optimal attacks across various dimensionalities. We filter out samples that have an $\ell_2$ norm less than $\epsilon$ to ensure meaningful perturbations. One-sample t-tests comparing observed similarities against the random baseline yield $p < 10^{-10}$ for all configurations, demonstrating that PGD attacks systematically match theoretical predictions rather than occurring by chance. Given that optimal attacks leverage interference and PGD attacks achieve near-perfect alignment with these optima, we conclude that PGD exploits superposition interference.

> **Finding**: Adversarial attacks systematically exploit interference between superposed features. Successful PGD attacks are predictable given the specific superposition geometry, rather than random.

**Do input correlations determine latent feature geometry?** We now investigate whether correlations in the input data determine the geometric arrangements of latent features. By systematically manipulating correlation structure in training data, we test how these correlations constrain the degrees of freedom in learned representations. We introduce three correlation patterns to vary the constraint strength. Under the *uncorrelated* condition, features are sampled independently as in the previous subsection. For *paired* correlations each feature in class $i$ is coupled with a feature in class $j$, such that when one activates, so does its pair. For *global* correlations, each sample has a random phase that determines a cyclic activation pattern: classes adjacent in index have similar activation probabilities and thus frequently co-activate, while classes further separated rarely activate together.

Fig. 2 displays how these correlations determine the arrangement of $\mathbf{v}^{(j)}$. We quantify geometric similarity by comparing the pairwise cosine similarity matrices between all feature pairs in each model, then measuring the correlation between these matrices across different random seeds. Higher correlation indicates more similar geometric arrangements. For each correlation condition, we tested 25 seed pairs per condition. For the $m = 2, k = 7$ setup, geometric similarity was $0.23 \pm 0.14$ (uncorrelated), $0.47 \pm 0.32$ (paired), and $0.88 \pm 0.06$ (global) (mean $\pm$ std). The results show a monotonic relationship: uncorrelated data yields highly variable geometries across seeds, paired correlations partially constrain the arrangements, whilst global correlations force near-identical geometries. Statistical significance was as-

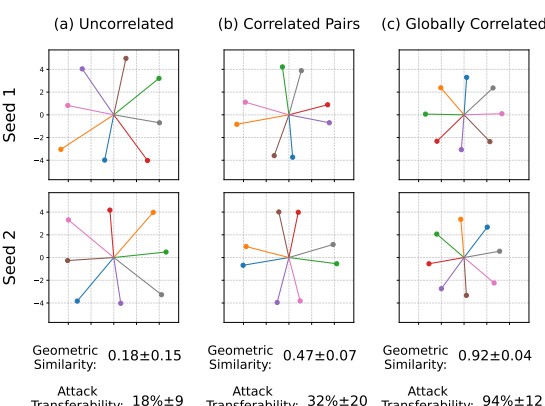

Figure 2: Greater input correlations create more consistent geometries between initialisations, driving attack transferability from 18% (uncorrelated) to 94% (global). Error bars show standard deviations.

sessed using pairwise two-sample t-tests between all three modes with Bonferroni correction ($p < 10^{-3}$ for all comparisons, $\alpha = 0.05$ corrected for 3 comparisons). Results across various $(k, m)$ configurations are provided in App. C.3.

The mechanism is *interference avoidance*: frequently co-activating features are arranged to minimise mutual interference. Stronger correlations impose more constraints: uncorrelated data allows any arrangement that separates classes, paired correlations fix relative positions of coupled features, and global correlations determine a unique geometry up to rotation.

> **Finding:** Input correlations constrain feature geometry. Stronger correlations reduce geometric degrees of freedom, forcing different initialisations to converge to similar arrangements.

**Does shared geometry explain attack transferability?** Having established that input correlations induce consistent latent geometries across model initialisations, we now test whether the similar interference patterns explains attack transferability. If attacks are tailored for specific interference patterns, they should transfer successfully only between models with similar geometric arrangements.

Using the same seed pairs as the previous paragraph, we generate adversarial attacks on source models and evaluate their success when applied to target models within each correlation condition (25 transfer measurements per condition). All models achieve $> 95\%$ clean accuracy. Transfer rates correlate strongly with geometric similarity. For the $k = 7, m = 2$ configuration (Fig. 2): globally correlated data yields $94\% \pm 12\%$ attack transfer (mean $\pm$ std), uncorrelated data shows only $18\% \pm 9\%$ transfer, and paired correlations produce intermediate results of $32\% \pm 20\%$ transfer. Results across various $(k, m)$ configurations are provided in Appendix A.1.3.

Each perturbation component amplifies or suppresses features, creating constructive interference that pushes representations across decision boundaries. When the same perturbation is applied to a model with different feature arrangements, features that previously constructively interfered instead cancel out or interfere destructively, causing the attack to fail.

> **Finding:** Attack transferability is governed by shared interference patterns.

**Does reducing superposition suppress these attacks?** If superposition creates vulnerability through interference, then removing superposition should limit adversarial vulnerability. We test this via three experiments that limit interference. First we set $m = k$, and the network learns to represent each class direction $\mathbf{v}^{(j)}$ orthogonally. In this configuration, any perturbation that changes the model's prediction also changes the ground truth class. To move a sample from class A to class B requires making the sum of class B features exceed class A's—genuinely transforming it into a class B sample. We find zero successful adversarial examples across 1000 attempts at all $\epsilon$ values tested. Second, we isolate superposition's effect from capacity by fixing $m$ and varying $k$. Robust accuracy decreases monotonically with superposition pressure $k/m$ (App. Tab. 5), demonstrating that vulnerability scales with interference degree. Thirdly, we test whether attacks focus on superposed features by constraining one class's latent vector $\mathbf{v}^{(\perp)}$ to remain orthogonal to all others during training. When generating attacks between classes that remain in superposition (*e.g.* from class $a$ to $b$ where $\mathbf{v}_a \not\perp \mathbf{v}_b$), the inputs corresponding to the orthogonal class remain unperturbed (Fig. 3).

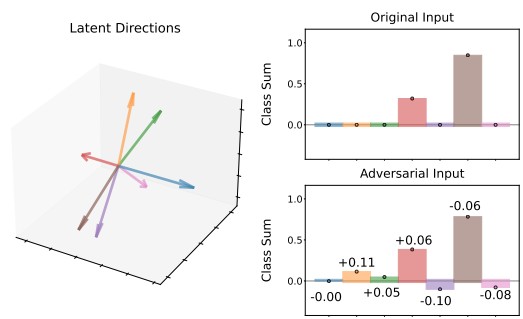

Figure 3: An adversarial attack (from class 5 to class 3) does not perturb the input features for a class represented orthogonally (class 1).

> **Finding:** Adversarial attacks use their budget to exploit those features in superposition.

## 3.2 FORMAL ANALYSIS

We now formalise the mechanisms underlying our empirical observations. In our linear setting, we can derive exact relationships between superposition interference and adversarial vulnerability, extending our results beyond observational correlation. We adopt the notation from our experimental setup, where $\mathbf{W}_e \in \mathbb{R}^{m \times d}$ induces superposition with columns $\mathbf{v}_i$, and the decoder corresponds to the encoder transposed (as empirically observed). The logit for class $j$ is $z_j = \mathbf{v}_j^\top \mathbf{h}$.

**Proposition 1.** *The optimal input perturbations $\boldsymbol{\delta}$ that maximise movement from class $j$ to class $k$ under constraint $\|\boldsymbol{\delta}\|_2 = \epsilon$ satisfy $\boldsymbol{\delta} \propto \mathbf{W}_e^\top \mathbf{n}$, where $\mathbf{n} = (\mathbf{v}_k - \mathbf{v}_j)$ is the normal to the decision boundary between classes.*

*Proof Sketch.* To move from class $j$ to $k$, we maximise the logit margin $z_k - z_j = \mathbf{n}^\top (\mathbf{h} + \Delta\mathbf{h})$, where $\mathbf{n} = (\mathbf{v}_k - \mathbf{v}_j)$ and $\Delta\mathbf{h} = \mathbf{W}_e \boldsymbol{\delta}$. This becomes $\max_{\|\boldsymbol{\delta}\|_2 = \epsilon} \boldsymbol{\delta}^\top \mathbf{W}_e^\top \mathbf{n}$. By Cauchy-Schwarz, this is maximised when $\boldsymbol{\delta} \propto \mathbf{W}_e^\top \mathbf{n}$. Full proof in App. B. $\qquad\square$

**Corollary 1 (Interference Drives Vulnerability)** *The adversarial perturbation magnitude for feature $i$ is $|\delta_i| \propto |\mathbf{v}_i^\top (\mathbf{v}_k - \mathbf{v}_j)|$, directly proportional to the differential interference between feature $i$ and the class representations.*

This reveals how superposition creates adversarial vulnerability. Each input feature $i$ is perturbed proportionally to its interference with the class representations. Under superposition, the non-

orthogonality means semantically unrelated features interfere with the class decision—adversarial perturbations exploit these dependencies to manipulate outputs.

**Proposition 2.** *Models with feature representations related by orthogonal transformation* $\mathbf{Q}$ *(where* $\mathbf{Q}^\top \mathbf{Q} = \mathbf{I}$*) share identical optimal attack directions in input space.*

*Proof Sketch.* Under transformation $\mathbf{v}'_i = \mathbf{Q}\mathbf{v}_i$, optimal perturbations remain invariant: $\delta'_i \propto (\mathbf{v}'_i)^\top(\mathbf{v}'_k - \mathbf{v}'_j) = \mathbf{v}_i^\top \mathbf{Q}^\top \mathbf{Q}(\mathbf{v}_k - \mathbf{v}_j) = \mathbf{v}_i^\top(\mathbf{v}_k - \mathbf{v}_j) = \delta_i$. Full proof in App. B. $\square$

> Together with our empirical findings, these propositions establish a **mechanistic pathway**. Specifically: (i) input correlations constrain feature arrangements in superposition, (ii) these geometric arrangements determine interference patterns, (iii) interference patterns dictate optimal perturbations via $\delta \propto \mathbf{W}_e^\top(\mathbf{v}_k - \mathbf{v}_j)$ (Proposition 1), and (iv) shared geometric constraints yield similar interference patterns across models, enabling transferability (Proposition 2):
>
> **Correlations** $\xrightarrow{\text{constrain}}$ **Feature Geometry** $\xrightarrow{\text{determines}}$ **Interference Patterns** $\xrightarrow{\text{enable}}$ **Transferability**

## 4 ATTACKS IN REALISTIC MODELS

We now extend our analysis to more realistic models, namely a ViT (Dosovitskiy et al., 2020) trained on CIFAR-10 (Krizhevsky, 2009) (and ResNet-18 (He et al., 2016) App. D.2.1). By introducing a bottleneck layer directly before the final classification output, we again force the latent class features to be represented in superposition. We then examine: (i) how the degree of superposition impacts attack transferability and robustness, (ii) if a consistent arrangement of the superposed representations emerges across initialisations, and (iii) if the attacks exploit feature interference.

**Setup**. We train our ViT architecture (6 layers, patch size 4, residual stream $d = 512$) on CIFAR-10 across five random seeds. After training, we replace the original classification head with a bottleneck consisting of a linear encoder that compresses the pre-classification activations to an $m$-dimensional space, followed by a linear decoder that expands back to the $k = 10$ CIFAR-10 classes. This bottleneck structure, trained with CE loss and no ReLU (mirroring the setup in Sec. 3), forces the $k$ class representations into $m$-dimensional superposition. During training of the bottleneck layer the weights of the base ViT are frozen. We vary the bottleneck dimension $m \in \{2, 3, 5, 10\}$ to control the degree of superpositional compression. AExs are generated for each bottlenecked model using both $\ell_\infty$- and $\ell_2$-norm PGD. We employ a small step size and a large number of iterations, terminating when an attack successfully changes the model's prediction (full details in App. D.1). Robust accuracy is normalised by the model's clean accuracy to account for performance variations due to different bottleneck capacities. Attack transferability is measured as defined in Sec. 3.

**Results**. The mean clean accuracy of the base models is 81%. As expected, the clean accuracy of the bottlenecked models decreases with compression: 54%, 67%, 76%, and 81% for $m = \{2, 3, 5, 10\}$.

We make three observations that mirror our toy-model results, evidencing transfer to this more realistic setting. First, Fig. 4 (right) shows that as the bottleneck $m$ decreases (*i.e.* superposition pressure increases): (i) normalised robust accuracy decreases, and (ii) attack transferability across different model initialisations increases. The decrease in robust accuracy with increasing superposition is likely due to the denser packing of class representations, making it easier for small input perturbations to cause widespread interference across these compressed features. The increased transferability follows from our Sec. 3 findings that higher superposition reduces the degrees of freedom in potential feature geometry. With a more constrained representational space available, the network has fewer viable geometric arrangements for its class features. This leads to different model initialisations converging to more similar representational geometries and, consequently, more shared interference patterns, which results in greater attack transferability. See App. D.1.1 for results across perturbation magnitudes.

Supporting this, Fig. 4 (left) shows that the geometric arrangement of the $m$-dimensional class representations within the bottleneck exhibit consistent ordering and clustering across models. This consistency, akin to findings in Sec. 3, suggests that inherent correlations guide the formation of a superposed geometry. Notably, these groupings often reflect semantic similarities between classes (*e.g.* 'cat' and 'dog' representations clustering together). Finally, when AExs generated for a bottlenecked model are run through the corresponding base ViT, they produce similar relative logit changes (*i.e.*

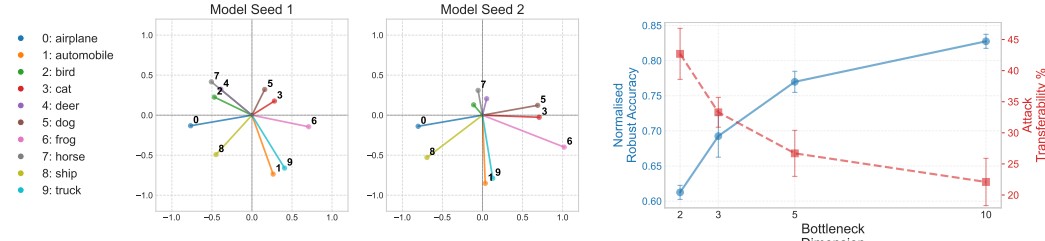

Figure 4: Left: CIFAR-10 class representation structure remains similar between models across different seeds. Right: Attack transferability and robust accuracy as bottleneck dimension is increased.

$\text{logit}_{base}(\mathbf{x}_{adv}) - \text{logit}_{base}(\mathbf{x}_{orig}))$. This implies both models respond consistently to the same perturbations. The perturbations are not solely an artefact of the bottleneck. However, without the increased compression these do not lead to misclassifications.

## 5  ADVERSARIAL VULNERABILITY VIA ALGORITHMIC BRITTLENESS

Prior sections argued that feature superposition can *suffice* for adversarial vulnerability. Here we investigate whether it is *necessary* by studying a setting where features are represented orthogonally, not in superposition, yet remain vulnerable to attacks. This supports a broader view developed in the paper: the analysis of feature representations can be used in robustness analysis. We ask three questions: *how do such attacks work mechanistically*, *can this understanding be used to construct attacks directly, without gradients*, and *does naive robustness certification alter the underlying failure mode or merely increase required attack budgets*?

**Setup**. We train a linear encoder + MLP on one-hot integers $(a, b)$ to predict $(a + b) \mod P$. As detailed in App. E and Nanda et al. (2023), the model discovers a trigonometric algorithm to solve this task. As part of this learned algorithm[1] the encoder projects inputs onto non-basis-aligned linear representations encoding $\sin(\omega_k a)$ and $\cos(\omega_k a)$ at key frequencies $\omega_k = 2\pi k / P$. Ultimately, the MLP composes $\cos((a + b) \omega_k)$ via sum-angle identities to obtain 100% accuracy. Crucially, these features are represented *orthogonally* (*i.e.* not in superposition). We generate AExs by relaxing the one-hot constraint and perturbing only input $a$ using PGD, moving inputs off the training manifold.

**Attacks exploit the learned key frequencies**. Fourier analysis of successful perturbations $\boldsymbol{\delta}$ (App. Fig. 6) reveal that dominant frequencies precisely match the learned algorithm's key frequencies, $\omega_k$. 0% of attacks transfer to models with different $\omega_k$, *e.g.* those trained from different seeds. The frequency-aligned perturbations cause misclassification by effectively altering the sinusoidal values read off from the encoder features. The perturbation $\boldsymbol{\delta}$ transforms one-hot $\mathbf{x}_a$ into a dense vector $\mathbf{x}_a'$. The introduced contributions from the now non-zero inputs corrupt the sinusoidal values, thereby disrupting the MLP's subsequent computations and leading to misclassification. As the model's algorithm uses the constructive interference of each $\omega_k$ at the logits, disrupting a single $\omega_k$ is insufficient. Successful attacks must find $\boldsymbol{\delta}$ that maximally alters multiple frequency components. We leave a detailed intuition of the mechanism to App. E.1

**Analytic construction of attacks**. Using this mechanism, we can construct gradient-free, *informed* perturbations formed from composite sinusoidal waves of matching $\omega_k$. On $0 + 0 \mod P$, these attacks succeed with $\|\boldsymbol{\delta}\|_2 = 0.22$, close to PGD ($0.17$, $\approx 1.3\times$ the PGD budget) and far below naive alternatives (single-index 0.97, $\ell_\infty$ noise 2.04, uniform 6.20). This provides empirical evidence that we are attacking the correct mechanism and shows that a representation-aware view of vulnerability can yield concrete attack recipes rather than merely post-hoc explanations.

**Do defences mitigate this algorithmic vulnerability?** We apply certified training against $\ell_\infty$ perturbations (Müller et al., 2023). For fixed attack budgets, the *robust* accuracy improves from pre-robustified $\to$ post-robustified as follows: $0\% \to 85\%$ at $\epsilon_\infty = 5e{-}3$, $0\% \to 96\%$ at $2e{-}3$, and $76\% \to 100\%$ at $1e{-}3$. However, when attacked with PGD, the robust model still fails via the same frequency-based brittleness, but requires larger perturbation norms. Certification therefore raises

---

[1]We refer to the specific computation executed by the network to solve the task as the *model's learned algorithm*. As this algorithm depends on extracting precise values from learned feature representations (frequencies), it remains *brittle* in the face of small perturbations to the bases. We term this the *algorithmic brittleness*.

budgets without addressing the specific fragility itself, motivating defences that explicitly target identified semantics (here, frequency manipulation) rather than relying solely on broad norm bounds.

> **Relation to the paper's thesis.** This section makes two points that integrate with the paper's broader narrative. First, superposition is *sufficient but not necessary*: orthogonal representations can still be vulnerable to attack. Secondly, viewing AExs through the lens of *architecture–semantics interaction* is actionable: it can guide both the design of attacks (via representation-informed perturbations) and the development of defences.

Although less central here, we note a shared dependence on sparsity assumptions: upstream violations of sparsity (*e.g.* noise or superposed features) can deliver effectively 'pre-perturbed' inputs to downstream orthogonal circuits, activating the same frequency-based failure.

## 6    RELATED WORK

**Superposition & latent representations.** Superposition research identifies feature interference mechanisms: Nanda (2023) distinguish *representational* and *computational* superposition; Gurnee et al. (2023) differentiate *alternating* and *simultaneous* interference. Data correlations shape arrangements: pairs of correlated features become orthogonal (Elhage et al., 2022), drive superposition formation (Chan, 2024), and create semantic clustering (Prieto et al., 2025). Beyond superposition, representation geometry is shaped by: spectral bias (Rahaman et al., 2019), neural collapse (Kothapalli, 2023), and optimisation objectives (Casper, 2023). Sparse autoencoders (SAEs) have been applied to vision models to disentangle features in superposition (Lim et al., 2024; Joseph et al., 2025).

**Adversarial vulnerability.** The literature follows an attack-defence dichotomy. Attacks (Szegedy et al., 2014) use imperceptible perturbations to maximize model error, including FGSM (Goodfellow et al., 2014) and PGD (Madry et al., 2018). Adversarial robustness describes maintaining performance under attacks. Defenses include adversarial training (Madry et al., 2018) and certified training (Wong & Kolter, 2018; Palma et al., 2024) providing guarantees against perturbations within input bounds.

**The role of input & latent features on adversarial robustness.** Vulnerability is attributed to predictive but brittle 'non-robust' input features (Ilyas et al., 2019; Goh, 2019), or learning shortcuts (Li et al., 2023). Perturbation characteristics vary by dataset (Maiya et al., 2021). Elhage et al. (2022) suggest that superposition may link to AExs, though Casper (2023) debates this. A tension between representational efficiency and adversarial robustness has been observed (Barsbey et al., 2025). Zhang et al. (2021) show adversarial perturbations push latent representations across decision boundaries. Ganeshan et al. (2019) find PGD targets final layers and propose feature-based attacks. Transferability is limited by representation discrepancies between models (Li et al., 2023; Wang et al., 2024); decorrelating features reduces transfer (Wiedeman & Wang, 2022), aligning with our findings.

## 7    DISCUSSION & CONCLUDING REMARKS

We demonstrate that adversarial attacks can exploit interference patterns arising from the geometry of superposed features in NNs. Our experiments, spanning toy models and ViTs, establish that data properties - namely correlations and sparsity - induce distinct superposition geometries, which in turn create predictable adversarial vulnerabilities. We show that these geometric arrangements and the resulting interference allow for the prediction of phenomena such as attack transferability and class-specific susceptibility. Our new perspective frames adversarial vulnerability as a potential, inherent consequence of how networks efficiently encode vast amounts of information via superposition.

**Limitations & future work**. Our insights stem primarily from simplified models. While our analyses offer an initial mechanistic lens for attacks in large models, the vast adversarial robustness literature suggests that precise vulnerability mechanisms likely vary by attack and model type. For example, constrained attack types (*e.g.* universal perturbations, jailbreaks) may be unable to activate the specific feature combinations required to exploit superposition interference.

Future work includes quantifying the 'adversarial cost' of superposition, its connection to accuracy-robustness trade-offs, and how robust training reshapes superposition geometry. Our insights derive from simplified settings with class features in engineered superposition, while real networks involve interference between unknown features across multiple layers. We plan to use SAEs (Bricken et al., 2023) to analyse networks, with preliminary results in App. F indicating this a promising direction.

## ETHICS STATEMENT

Due to its investigative nature and generality, our mechanistic interpretability framework is not anticipated to have any specific negative impact or immediate societal consequences. Our experiments utilise only publicly available datasets (CIFAR-10) and synthetic toy models, involving no human subjects or personally identifiable information. The gradient-based attack methods used in the paper are already well-established in the literature. Our work does not introduce novel attack capabilities but rather provides theoretical understanding of existing phenomena. We have no conflicts of interest to declare. By elucidating the relationship between superposition and adversarial vulnerability, we hope to inform future research directions that can balance model capability with robustness, ultimately contributing to more trustworthy AI systems that can be safely deployed in critical applications.

## REPRODUCIBILITY STATEMENT

We include source code for all experiments in the supplementary materials, which will be made public upon publication. The toy model experiments (Sec. 3) include complete architectural details, training procedures, evaluation metrics, and data generation processes. For ViT experiments (Sec. 4), architecture and training procedures are provided, with further details in App. D.1. The modular arithmetic experiments (Sec. 5) detail the network architecture, data encoding, and learned trigonometric algorithm, with additional details in App. E.2. Experiments were conducted on an AMD Ryzen Threadripper 3970X 32-Core with 256GB RAM and RTX 3090. Additional ResNet-18 and SAE experiments are documented in Appendices App. D.2.1 and App. F, respectively.

## LLM USAGE STATEMENT

This work utilised LLMs to assist with prior literature search, automate LaTeX table generation from CSV files (all numerical values were verified), and suggest alternative phrasings to improve manuscript clarity. All substantive research contributions and conclusions remain the original work of the authors.

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

APPENDIX

## A   GENERATION OF ADVERSARIAL EXAMPLES

We use PGD to generate AExs, which is an iterative method to generate attacks (Madry et al., 2018). For untargeted attacks, it maximises the classifier's loss function $\mathcal{L}(f(\mathbf{x}'), y_{\text{true}})$; for targeted, it minimises $\mathcal{L}(f(\mathbf{x}'), y_{\text{target}})$ by the update rule:

$$\mathbf{x}'^{(k+1)} = \Pi_{\mathcal{S}}\left(\mathbf{x}'^{(k)} \pm \alpha \mathbf{g}_k\right) \tag{1}$$

(+ for maximisation, − for minimisation). $\mathbf{x}'^{(0)}$ is an initial perturbed input, $\alpha$ is step size, $\mathbf{g}_k$ is the (normalised) gradient $\nabla_{\mathbf{x}}\mathcal{L}(f(\mathbf{x}'^{(k)}), y_{\text{class}})$ ($y_{\text{class}}$ being $y_{\text{true}}$ or $y_{\text{target}}$), and $\Pi_{\mathcal{S}}$ projects onto the $\epsilon$-ball $\mathcal{S} = \{\mathbf{x}' \mid \|\mathbf{x}' - \mathbf{x}\|_p \leq \epsilon\}$, with input domain clipping. We use the $\ell_p$ norms:

1. $\ell_\infty$: Constraint $\|\boldsymbol{\delta}\|_{\ell_\infty} = \max_i |\boldsymbol{\delta}_i| \leq \epsilon$. $\mathbf{g}_k$ is $\text{sign}(\nabla_{\mathbf{x}}\mathcal{L}(\cdot))$. $\Pi_{\mathcal{S}}$ clips each $\mathbf{x}'_i$ to $[\mathbf{x}_i - \epsilon, \mathbf{x}_i + \epsilon]$.
2. $\ell_2$: Constraint $\|\boldsymbol{\delta}\|_{\ell_2} = \sqrt{\sum_i \boldsymbol{\delta}_i^2} \leq \epsilon$. $\mathbf{g}_k$ is $\nabla_{\mathbf{x}}\mathcal{L}(\cdot)/\|\nabla_{\mathbf{x}}\mathcal{L}(\cdot)\|_{\ell_2}$. $\Pi_{\mathcal{S}}$ rescales $\boldsymbol{\delta} = \mathbf{x}' - \mathbf{x}$ if $\|\boldsymbol{\delta}\|_{\ell_2} > \epsilon$ via $\boldsymbol{\delta} \leftarrow \epsilon \cdot \boldsymbol{\delta}/\|\boldsymbol{\delta}\|_{\ell_2}$.

## B   OMITTED PROOFS & THEORETICAL RESULTS

To understand how adversarial attacks exploit feature representations, we prove that optimal perturbations weight each input dimension by how much its corresponding feature aligns with the path to the decision boundary. We analyse linear models without activations.

Consider input $\mathbf{x} \in \mathbb{R}^d$ encoded via $\phi(\mathbf{x}) = \mathbf{W}_e\mathbf{x}$ to latent representation $\mathbf{h} \in \mathbb{R}^m$, where the columns of $\mathbf{W}_e \in \mathbb{R}^{m \times d}$ are overcomplete basis vectors $\{\mathbf{v}_i\}_{i=1}^d$ ($m < d$). For our argmax task, we empirically observe that the trained encoder-decoder pair has the decoder as the transpose of the encoder: $\mathbf{W}_d = \mathbf{W}_e^\top$. This means the logit for class $j$ is computed as $z_j = \mathbf{v}_j^\top\mathbf{h}$, where $\mathbf{v}_j$ is the $j$-th column of $\mathbf{W}_e$. The predicted class is $\hat{y} = \arg\max_j z_j$. Input perturbations map to latent perturbations as $\Delta\mathbf{h} = \mathbf{W}_e\boldsymbol{\delta}$, where $\boldsymbol{\delta} = (\delta_1, \ldots, \delta_d)^\top$ are the perturbation coefficients.

The decision boundary between classes $j$ and $k$ is the set of points where their logits are equal: $\mathcal{B}_{jk} = \{\mathbf{h} : z_j = z_k\}$, or equivalently $\{\mathbf{h} : (\mathbf{v}_k - \mathbf{v}_j)^\top\mathbf{h} = 0\}$. The vector $\mathbf{n} = (\mathbf{v}_k - \mathbf{v}_j)$ acts as the normal to this pairwise decision boundary, pointing in the direction that maximally increases the margin $z_k - z_j$. We briefly recall the propositions before providing corresponding proofs.

**Proposition 1 (Optimal targeted attack).** The optimal input perturbations $\boldsymbol{\delta}$ that maximise movement from class $j$ toward class $k$ under constraint $\|\boldsymbol{\delta}\|_2 = \epsilon$ satisfy:

$$\boldsymbol{\delta} \propto \mathbf{W}_e^\top\mathbf{n}$$

*Proof.* To move a sample from being classified as $j$ to being classified as $k$, we need to maximise the logit margin $z_k - z_j$. Under perturbation $\boldsymbol{\delta}$, the new margin becomes:

$$z'_k - z'_j = \mathbf{v}_k^\top(\mathbf{h} + \Delta\mathbf{h}) - \mathbf{v}_j^\top(\mathbf{h} + \Delta\mathbf{h}) \tag{2}$$

$$= (\mathbf{v}_k - \mathbf{v}_j)^\top(\mathbf{h} + \mathbf{W}_e\boldsymbol{\delta}) \tag{3}$$

$$= (\mathbf{v}_k - \mathbf{v}_j)^\top\mathbf{h} + (\mathbf{v}_k - \mathbf{v}_j)^\top\mathbf{W}_e\boldsymbol{\delta} \tag{4}$$

The change in margin is $(\mathbf{v}_k - \mathbf{v}_j)^\top\mathbf{W}_e\boldsymbol{\delta} = \boldsymbol{\delta}^\top\mathbf{W}_e^\top(\mathbf{v}_k - \mathbf{v}_j)$. We seek:

$$\max_{\boldsymbol{\delta}} \boldsymbol{\delta}^\top\mathbf{W}_e^\top(\mathbf{v}_k - \mathbf{v}_j) \quad \text{subject to} \quad \|\boldsymbol{\delta}\|_2 = \epsilon$$

Let $\mathbf{g} = \mathbf{W}_e^\top(\mathbf{v}_k - \mathbf{v}_j)$. By the Cauchy-Schwarz inequality:

$$|\boldsymbol{\delta}^\top\mathbf{g}| \leq \|\boldsymbol{\delta}\|_2\|\mathbf{g}\|_2 = \epsilon\|\mathbf{g}\|_2$$

Equality is achieved when $\boldsymbol{\delta}$ and $\mathbf{g}$ are parallel. Given the constraint $\|\boldsymbol{\delta}\|_2 = \epsilon$:

$$\boldsymbol{\delta} = \frac{\epsilon}{\|\mathbf{W}_e^\top(\mathbf{v}_k - \mathbf{v}_j)\|_2}\mathbf{W}_e^\top(\mathbf{v}_k - \mathbf{v}_j)$$

$\square$

**Corollary (Interference drives vulnerability).** For a targeted attack from class $j$ to class $k$, the adversarial perturbation magnitude for input feature $i$ is:

$$|\delta_i| \propto |\mathbf{v}_i^\top (\mathbf{v}_k - \mathbf{v}_j)|$$

where $\mathbf{v}_i$ is the $i$-th column of $\mathbf{W}_e$. This quantity represents the differential interference between feature $i$ and the class representations.

This reveals the mechanism by which superposition creates adversarial vulnerability. Each input feature $i$ is perturbed proportionally to $\mathbf{v}_i^\top (\mathbf{v}_k - \mathbf{v}_j)$—the differential interference between feature $i$ and the class representations. Under superposition, the non-orthogonality means that even semantically unrelated features have non-zero inner products with $(\mathbf{v}_k - \mathbf{v}_j)$, creating exploitable interference patterns. The multi-class setting amplifies this vulnerability, as with $k$ classes there are multiple possible pairwise boundaries, each creating a distinct interference pattern. Adversarial perturbations leverage these cross-feature dependencies—they manipulate features that affect it through their interference with the class representations. This explains the vulnerability we observe empirically: attacks succeed not by directly increasing target class features, but by exploiting the web of interference created by superposition.

Prop. 1 characterises the optimal perturbation direction for moving from class $j$ to $k$, providing the gradient for maximising the margin $z_k - z_j$. When intervening classes exist (where $z_j < z_i < z_k$ for some class $i$), following this gradient might cause the model to predict $i$ before reaching target $k$. Iterative methods like PGD handle this by recomputing gradients at each step—the global attack path emerges from repeated local decisions rather than a single optimisation. Furthermore, while our analysis assumes $\mathbf{W}_d = \mathbf{W}_e^\top$ based on our empirical observations, the framework extends to arbitrary decoders. In the general case, with decoder $\mathbf{W}_d \in \mathbb{R}^{k \times m}$ having rows $\mathbf{w}_j^\top$, the optimal perturbation becomes $\boldsymbol{\delta} \propto \mathbf{W}_e^\top (\mathbf{w}_k - \mathbf{w}_j)$, capturing interference between encoder directions and decoder weights.

**Proposition 2 (Attack transferability).** Consider encoders $\phi$ and $\psi$ with basis matrices $\mathbf{W}_e \in \mathbb{R}^{m \times d}$ and $\mathbf{W}_e' \in \mathbb{R}^{m \times d}$ whose columns are related by orthogonal transformation $\mathbf{v}_i' = \mathbf{Q}\mathbf{v}_i$ (where $\mathbf{Q}^\top \mathbf{Q} = \mathbf{I}$). If both models use their encoder transpose as decoder (i.e., $\mathbf{W}_d = \mathbf{W}_e^\top$ and $\mathbf{W}_d' = (\mathbf{W}_e')^\top$), then both models have identical optimal input perturbation vectors for any targeted attack from class $j$ to class $k$.

*Proof.* For model $\phi$, the optimal perturbation from Prop. 1 is:

$$\boldsymbol{\delta}^\phi \propto \mathbf{W}_e^\top (\mathbf{v}_k - \mathbf{v}_j)$$

For model $\psi$ with transformed columns $\mathbf{v}_i' = \mathbf{Q}\mathbf{v}_i$:

$$\boldsymbol{\delta}^\psi \propto (\mathbf{W}_e')^\top (\mathbf{v}_k' - \mathbf{v}_j') = (\mathbf{W}_e')^\top (\mathbf{Q}\mathbf{v}_k - \mathbf{Q}\mathbf{v}_j)$$

Since $\mathbf{W}_e' = \mathbf{Q}\mathbf{W}_e$ (all columns are transformed), we have:

$$\boldsymbol{\delta}^\psi \propto (\mathbf{Q}\mathbf{W}_e)^\top \mathbf{Q}(\mathbf{v}_k - \mathbf{v}_j) \tag{5}$$

$$= \mathbf{W}_e^\top \mathbf{Q}^\top \mathbf{Q}(\mathbf{v}_k - \mathbf{v}_j) \tag{6}$$

$$= \mathbf{W}_e^\top (\mathbf{v}_k - \mathbf{v}_j) \tag{7}$$

where the last equality uses $\mathbf{Q}^\top \mathbf{Q} = \mathbf{I}$. Thus $\boldsymbol{\delta}^\phi$ and $\boldsymbol{\delta}^\psi$ have identical proportionality. Under the same norm constraint $\|\boldsymbol{\delta}\|_2 = \epsilon$, we have $\boldsymbol{\delta}^\phi = \boldsymbol{\delta}^\psi$. $\square$

Prop. 2 is used to explain why attacks transfer between models with similar training regimes. When models learn feature representations that differ only by orthogonal transformation—essentially the same geometric structure in different orientations—they share identical vulnerability patterns in input space. The orthogonal transformation preserves all inner products between features, maintaining the interference patterns that attacks exploit. This provides an explanation for our empirical observation that models trained on data with the same correlations exhibit high attack transferability: they discover similar feature geometries up to rotation, leading to shared adversarial vulnerabilities.

## C  Toy Model Experiments

This section provides supplementary details, extended results and further discussion for the toy model experiments discussed in the main paper. We present model accuracies across a wider range of parameters than shown in the main text, offering insight into how model capacity and data characteristics like sparsity influence the learning process and the conditions under which feature superposition appears. Subsequently, we offer additional visual examples that correspond to Fig. 1, illustrating the mechanics of adversarial attacks under various conditions.

### C.1  Hypotheses Testing Framework

We explicitly state our hypotheses for the three research questions in Sec. 3.

*Research Q1*: Do adversarial perturbations exploit superposition geometry?

- $H_0$: Adversarial perturbations are random with respect to feature geometry.
- $H_1$: Adversarial perturbations systematically exploit geometric relationships between superposed representations.

*Research Q2*: Do data correlations determine superposition geometry?

- $H_0$: Input correlations have no systematic effect on learned geometries.
- $H_1$: Input correlations determine geometric arrangements across model initializations.

*Research Q3*: Does shared geometry explain attack transferability?

- $H_0$: Attack transferability is independent of geometric similarity.
- $H_1$: Transferability increases with shared latent structure.

We test these hypotheses through controlled experiments:

- $H_1(1)$: We measure the input perturbation profile alignment with a class's latent representation and the latent attack vector.
- $H_1(2)$: We systematically vary input correlations and measure resulting geometries.
- $H_1(3)$: We quantify transferability rates across models with varying geometric similarity.

### C.2  Accuracy of Toy Model for a Range of Parameters

The toy model experiments presented in the main paper predominantly used low-dimensionality settings for conceptual clarity. To demonstrate the model's behaviour more broadly, this subsection details the classification accuracies achieved by the CE toy model. These results are presented across varying hidden layer size ($h$), number of classes ($k$), number of features, and levels of sparsity ($S$), to provide insight into when the models learn to represent features in superposition. The sparsity level represents the probability that any individual input feature $x_i^{(j)}$ is set to zero, with higher values of $S$ indicating greater input sparsity. Tab. 2 and Tab. 3 provide provide context on the model's performance limits and its ability to learn latent representations in superposition.

### C.3  Correlations constrain arrangements

Tab. 4 displays the results for the experiments regarding *Do Input Correlations Determine Latent Feature Geometry?* across a wider range of classes ($k$) and hidden dimension size ($m$).

### C.4  How the toy model maps to real-world models

How scaling networks affects adversarial vulnerability? While empirical evidence suggests that larger models tend to be more robust to adversarial attacks, this effect is weak. When adversarial training is employed, clearer scaling trends emerge, but improvements remain largely specific to the attack type used during training rather than conferring general robustness (Howe et al., 2025).

There are two key phenomena at play when scaling up models. On one hand, larger models have more capacity to represent concepts, but on the other hand, there seems to be a long tail of useful concepts

Table 2: Classification accuracy of the CE toy model with a fixed bottleneck dimension ($m = 2$) across various numbers of classes ($k$), total input features (features = $k \times 3$), and input feature sparsity levels ($1 - S$). These results illustrate how input sparsity controls performance degradation as the number of classes to be superposed within a constrained latent space increases.

| Classes ($k$) | Features | Hidden ($m$) | Accuracy at Sparsity Level ($1 - S$) | | | | | | | |
|---|---|---|---|---|---|---|---|---|---|---|
| | | | 1.0 | 0.57 | 0.33 | 0.19 | 0.11 | 0.06 | 0.04 | 0.02 |
| 3 | 9 | 2 | 1.00 | 1.00 | 1.00 | 1.00 | 1.00 | 1.00 | 1.00 | 1.00 |
| 4 | 12 | 2 | 0.67 | 0.71 | 0.82 | 0.92 | 0.98 | 0.99 | 1.00 | 1.00 |
| 5 | 15 | 2 | 0.53 | 0.50 | 0.65 | 0.77 | 0.89 | 0.95 | 0.98 | 0.99 |
| 6 | 18 | 2 | 0.40 | 0.43 | 0.51 | 0.66 | 0.82 | 0.93 | 0.97 | 0.99 |
| 7 | 21 | 2 | 0.34 | 0.34 | 0.40 | 0.53 | 0.73 | 0.87 | 0.95 | 0.98 |
| 8 | 24 | 2 | 0.30 | 0.30 | 0.33 | 0.40 | 0.63 | 0.82 | 0.93 | 0.97 |
| 9 | 27 | 2 | 0.24 | 0.26 | 0.30 | 0.35 | 0.57 | 0.75 | 0.89 | 0.96 |
| 10 | 30 | 2 | 0.22 | 0.24 | 0.26 | 0.31 | 0.50 | 0.72 | 0.87 | 0.95 |
| 15 | 45 | 2 | 0.14 | 0.15 | 0.16 | 0.17 | 0.25 | 0.40 | 0.65 | 0.86 |
| 20 | 60 | 2 | 0.10 | 0.10 | 0.11 | 0.13 | 0.15 | 0.24 | 0.44 | 0.76 |
| 25 | 75 | 2 | 0.07 | 0.08 | 0.09 | 0.10 | 0.12 | 0.16 | 0.26 | 0.62 |
| 30 | 90 | 2 | 0.06 | 0.07 | 0.07 | 0.07 | 0.09 | 0.11 | 0.21 | 0.45 |

Table 3: Classification accuracy of the CE toy model for varying numbers of classes ($k$), total input features, bottleneck dimensions ($m$), and input feature sparsity levels ($1 - S$). This table reports different numbers of features per class ($p$).

| Classes ($k$) | Features | Hidden ($m$) | Accuracy at Sparsity Level ($1 - S$) | | | | | | | |
|---|---|---|---|---|---|---|---|---|---|---|
| | | | 1.0 | 0.57 | 0.33 | 0.19 | 0.11 | 0.06 | 0.04 | 0.02 |
| 30 | 30 | 30 | 0.23 | 0.24 | 0.38 | 0.62 | 0.83 | 0.94 | 0.99 | 1.00 |
| 30 | 90 | 90 | 0.27 | 0.27 | 0.33 | 0.41 | 0.51 | 0.67 | 0.85 | 0.94 |
| 40 | 40 | 30 | 0.67 | 0.54 | 0.73 | 0.77 | 0.88 | 0.96 | 0.99 | 0.99 |
| 40 | 120 | 30 | 0.71 | 0.64 | 0.65 | 0.72 | 0.73 | 0.79 | 0.89 | 0.96 |
| 60 | 60 | 10 | 0.05 | 0.07 | 0.12 | 0.25 | 0.47 | 0.73 | 0.90 | 0.97 |
| 60 | 180 | 10 | 0.08 | 0.10 | 0.13 | 0.17 | 0.22 | 0.32 | 0.52 | 0.75 |
| 80 | 80 | 30 | 0.15 | 0.17 | 0.23 | 0.41 | 0.63 | 0.79 | 0.91 | 0.98 |
| 80 | 240 | 30 | 0.23 | 0.22 | 0.31 | 0.41 | 0.48 | 0.53 | 0.66 | 0.80 |
| 100 | 100 | 10 | 0.03 | 0.04 | 0.05 | 0.10 | 0.21 | 0.43 | 0.69 | 0.87 |
| 100 | 300 | 10 | 0.04 | 0.05 | 0.06 | 0.09 | 0.12 | 0.15 | 0.25 | 0.45 |

Table 4: Geometric similarity results across correlation types and superposition configurations.

| Correlation Type | $k$ | $m$ | Geometric Similarity |
|---|---|---|---|
| Uncorrelated | 6 | 2 | $0.18 \pm 0.15$ |
| | 30 | 10 | $0.17 \pm 0.02$ |
| | 90 | 30 | $0.17 \pm 0.01Y$ |
| Paired | 6 | 2 | $0.47 \pm 0.07$ |
| | 30 | 10 | $0.26 \pm 0.07$ |
| | 90 | 30 | $0.25 \pm 0.01$ |
| Global | 6 | 2 | $0.92 \pm 0.04$ |
| | 30 | 10 | $0.88 \pm 0.01$ |
| | 90 | 30 | $0.80 \pm 0.01$ |

for a larger model to capture in general tasks like next token prediction over internet text. This means that despite increased model capacity, superposition appears to be prevalent even in frontier models (Lindsey et al., 2025). Supporting evidence comes from dictionary learning methods: SAEs require increasingly large dictionaries for larger models (Gao et al., 2025), suggesting that the number of features scales with model size. This suggests that the fundamental tension driving superposition – that models must compress many features into limited dimensions – does not disappear with scale.

Since both superposition and adversarial vulnerability persist in large-scale models, we believe our insights remain relevant across model scales. It is an interesting future avenue to understand how the geometry of superposition changes with scale, potentially helping to mitigate vulnerability.

### C.4.1 SEPARATING SUPERPOSITION EFFECTS FROM CAPACITY REDUCTION

Controlling for capacity by keeping the bottleneck dimension fixed to isolate superposition effects controls for the confounding effect of superposition pressure and capacity reduction. We fix bottleneck dimension $m = 2$, and vary number of classes $k$ to isolate superposition pressure. We report the model accuracy, robust accuracy, and transferability in Tab. 5. These controls demonstrate that increased superposition pressure ($k/m$), independent of model capacity, drives the adversarial vulnerabilities we observe.

Table 5: Performance metrics across correlation types, perturbation budgets ($\varepsilon$), and network depths ($k$)

| Correlation Type | Parameters | | Performance (%) | | |
|---|---|---|---|---|---|
| | $\varepsilon$ | $k$ | Accuracy | Robust Acc. | Attack Trans. |
| Uncorrelated | 0.05 | 2 | 100.0 | 100.0 | 0.0 |
| | | 4 | 100.0 | 94.9 | 21.2 |
| | | 6 | 99.9 | 87.0 | 9.8 |
| | | 8 | 99.8 | 77.7 | 17.5 |
| | 0.1 | 2 | 100.0 | 100.0 | 0.0 |
| | | 4 | 100.0 | 88.1 | 9.8 |
| | | 6 | 99.7 | 75.4 | 9.9 |
| | | 8 | 99.6 | 56.3 | 23.5 |
| | 0.5 | 2 | 100.0 | 100.0 | 0.0 |
| | | 4 | 100.0 | 99.4 | 0.0 |
| | | 6 | 99.9 | 62.9 | 30.5 |
| | | 8 | 99.5 | 24.9 | 39.0 |
| Fully Correlated | 0.05 | 2 | 100.0 | 100.0 | 0.0 |
| | | 4 | 100.0 | 95.8 | 98.4 |
| | | 6 | 97.3 | 70.6 | 98.4 |
| | | 8 | 93.0 | 57.4 | 97.2 |
| | 0.1 | 2 | 100.0 | 100.0 | 0.0 |
| | | 4 | 100.0 | 88.7 | 98.5 |
| | | 6 | 97.7 | 45.6 | 99.7 |
| | | 8 | 94.2 | 34.8 | 98.6 |
| | 0.5 | 2 | 100.0 | 100.0 | 0.0 |
| | | 4 | 100.0 | 90.0 | 100.0 |
| | | 6 | 97.2 | 34.6 | 100.0 |
| | | 8 | 93.0 | 32.0 | 100.0 |

### C.4.2 ADDITIONAL EXAMPLES OF AEXS IN TOY MODEL

Fig. 1 demonstrates how adversarial attacks exploit the interference between latent features in superposition. Here we provide further visual examples (Fig. 5) to reinforce intuition. Specifically, we supplement the main text by showcasing:

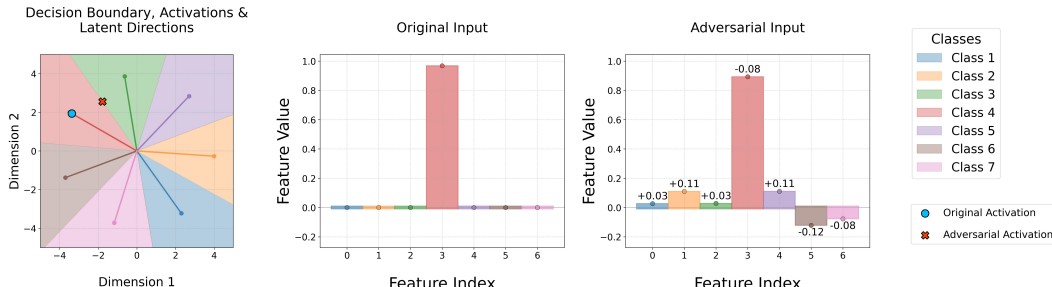

(a) An $\ell_2$-norm attack changing the classification of an input of class 4 to class 3. The left plot shows original and adversarial activations in latent space, along with representation directions. The right plots show original and perturbed input feature values.

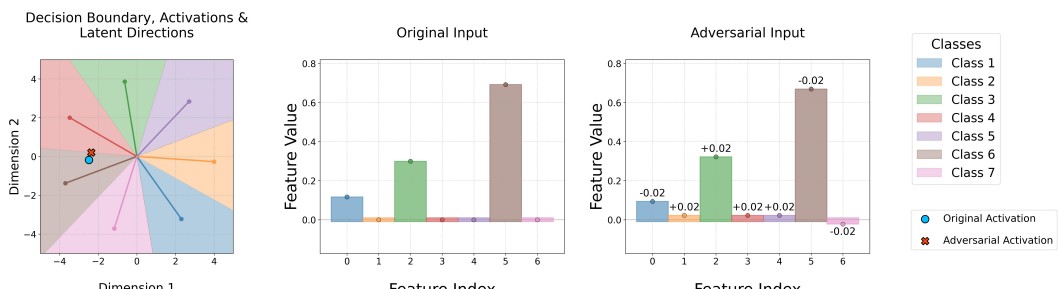

(b) An $\ell_\infty$-norm attack changing the classification of an input of class 6 to class 4. The left plot shows original and adversarial activations in latent space relative to class latent directions. The right plots show original and perturbed input feature values.

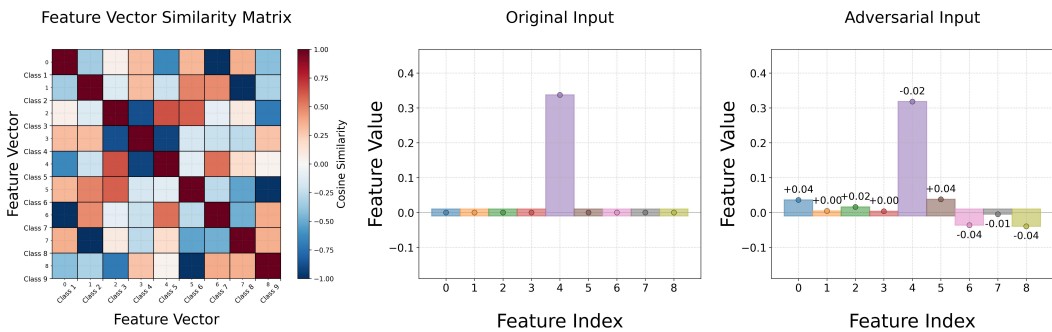

(c) An $\ell_2$-norm adversarial attack in a 7-class setup with an increased bottleneck dimension $m = 3$. The leftmost plot now shows the cosine similarity matrix between pairs of latent representations for each of the classes.

Figure 5: Visualisations of AExs in the toy model, supplementing Figure 1 from the main paper by illustrating attack mechanisms in activation space and input space under varied conditions.

- Fig. 5a shows an additional instance of the setup in Sec. 3.1 ($m = 2$, $k = 7$) with an $\ell_2$-norm PGD attack, demonstrating the IPP and latent space manipulations that lead to misclassification.
- Fig. 5b shows a similar setup ($m = 2$, $k = 7$) using an $\ell_\infty$-norm PGD attack on a less sparse input.
- Fig. 5c shows an example with increased bottleneck dimensionality ($m = 3$, $k = 7$) and accompanying $\ell_2$-norm PGD attack. The feature vector similarity matrix used to calculate geometric similarity is also shown.

# D  CIFAR-10 EXPERIMENTS

To investigate whether the principles observed in the toy models extend to more complex settings, Sec. 4 introduces experiments trained on CIFAR-10 (Krizhevsky, 2009) with an engineered bottleneck. This appendix section provides further details on this setup and presents extended results.

## D.1  ARCHITECTURE & TRAINING INFORMATION

The base ViT (Dosovitskiy et al., 2020) architecture comprised 6 transformer layers, an embedding dimension ($d$) of 512, and 8 attention heads in each transformer layer. Input images from the CIFAR-10 dataset, sized at $32 \times 32$ pixels, were processed into patches of $4 \times 4$ pixels. The Multilayer Perceptron (MLP) within each transformer block had a hidden dimension of 512. Learned positional embeddings were used.

The bottleneck architecture consisted of a linear encoder followed by a linear decoder. The linear encoder projected the pre-classification activations obtained from the ViT backbone into an $m$-dimensional latent space ($\{2, 3, 5, 10\}$). The subsequent linear decoder then mapped these $m$-dimensional representations back to the $k = 10$ dimensions corresponding to the CIFAR-10 classes.

The base ViT model was trained on the CIFAR-10 dataset for 250 epochs. A learning rate of 0.001 was used with the Adam optimiser using default PyTorch parameters. The batch size was set to 512 and a cosine annealing learning rate scheduler. The loss function was CE. Dropout was used. Training was performed across five different random seeds to account for variability. After the base ViT model was trained, its weights were frozen. The bottleneck layer was then trained for 30 epochs, utilising a learning rate of 0.001. As for preprocessing, the images undergo RandomCrop with 4-pixel padding, resize to the target size (32x32 by default), and RandomHorizontalFlip for data augmentation.

### D.1.1  NORMALISED ROBUST ACCURACY ACROSS PERTURBATION MAGNITUDES

Fig. 4(right) shows how normalised robust accuracy varies with bottleneck dimension for one value of $\epsilon$. This subsection expands on those findings by detailing the normalised robust accuracy when subjected to PGD attacks of different strengths. Results are presented for both $\ell_2$-norm (Tab. 7) and $\ell_\infty$-norm (Tab. 6) PGD attacks, providing a more comprehensive view of how the degree of superposition interacts with attack strength to affect model robustness.

AExs for these evaluations were generated using PGD with 100 iterations with a step size ($\alpha$) of 0.01. Robust accuracy was evaluated on 500 samples for each configuration. The mean normalised robust accuracy and standard deviation across five random seeds are reported.

Table 6: Mean normalised robust accuracy ($\pm$ standard deviation across 5 seeds) for ViT models with different bottleneck dimensions ($m$) on CIFAR-10, subjected to $\ell_\infty$-norm PGD attacks of varying perturbation magnitudes ($\epsilon$). Robust accuracy is normalised by the clean accuracy of each bottlenecked model. These results support the findings of Sec. 4, demonstrating a similar trend across $\epsilon$.

| $\epsilon$ | Bottleneck Dimension ($m$) | | | |
|---|---|---|---|---|
| | 2 | 3 | 5 | 10 |
| 0.001 | $96.0\% \pm 0.4\%$ | $97.4\% \pm 0.7\%$ | $97.9\% \pm 0.4\%$ | $98.1\% \pm 0.3\%$ |
| 0.01 | $61.7\% \pm 3.6\%$ | $69.6\% \pm 3.5\%$ | $77.0\% \pm 0.9\%$ | $81.8\% \pm 3.2\%$ |
| 0.05 | $4.9\% \pm 1.5\%$ | $6.7\% \pm 1.6\%$ | $9.3\% \pm 0.6\%$ | $10.5\% \pm 0.8\%$ |
| 0.1 | $0.1\% \pm 0.2\%$ | $0.2\% \pm 0.3\%$ | $0.2\% \pm 0.4\%$ | $0.2\% \pm 0.4\%$ |
| 0.5 | $0.0\% \pm 0.0\%$ | $0.0\% \pm 0.0\%$ | $0.0\% \pm 0.0\%$ | $0.0\% \pm 0.0\%$ |

Table 7: Mean normalised robust accuracy ($\pm$ standard deviation across 5 seeds) for ViT models with different bottleneck dimensions ($m$) on CIFAR-10, subjected to $\ell_2$-norm PGD attacks of varying perturbation magnitudes ($\epsilon$). Robust accuracy is normalised by the clean accuracy of each bottlenecked model. These results support the findings in Section 4 of the main paper, showing decreasing robustness with smaller $m$ (increased superposition) and larger $\epsilon$.

| $\epsilon$ | Bottleneck Dimension ($m$) | | | |
|---|---|---|---|---|
| | 2 | 3 | 5 | 10 |
| 0.1 | $90.4\% \pm 1.7\%$ | $91.8\% \pm 0.7\%$ | $94.6\% \pm 1.2\%$ | $95.0\% \pm 0.7\%$ |
| 0.5 | $58.5\% \pm 5.0\%$ | $60.8\% \pm 5.0\%$ | $69.2\% \pm 1.8\%$ | $72.6\% \pm 2.8\%$ |
| 1.0 | $41.7\% \pm 4.8\%$ | $44.0\% \pm 3.0\%$ | $50.4\% \pm 0.6\%$ | $54.9\% \pm 1.7\%$ |
| 2.0 | $34.0\% \pm 4.6\%$ | $36.2\% \pm 3.3\%$ | $41.5\% \pm 1.3\%$ | $47.4\% \pm 2.4\%$ |
| 5.0 | $30.2\% \pm 4.1\%$ | $32.9\% \pm 3.5\%$ | $37.3\% \pm 1.9\%$ | $42.7\% \pm 1.0\%$ |

### D.1.2 ATTACK TRANSFERABILITY ACROSS PERTURBATION MAGNITUDES

We here include results on attack transferability across various perturbation magnitudes ($\epsilon$), $\epsilon$ values and bottleneck dimensions ($m$). Table 9 presents the $\ell_2$-norm attack transferability and Table 8 $\ell_\infty$-norm attack transferability.

### D.2 CORRELATIONS IN CLASS FEATURES

In Sec. 3 it was correlations between inputs that drove superposition arrangements. We note that here it is not the correlations between input classes but rather the correlations in the representations at this point in the network that drive these arrangements. At the classification layer this is likely similar to the confusion matrix – *i.e.* how each class is misclassified in relation to the other classes. Classes that cluster together are those the network finds inherently similar and misclassifies together. To test this we repeat the experiment, finetuning the base ViT using timm/vit_base_patch16_384 which has been trained on ImageNet-21k (14 million images, 21,843 classes) and ImageNet (1 million images, 1,000 classes). After fine-tuning on CIFAR-10 and applying the same bottleneck training procedure as Sec. 4, the resulting geometry shows random ordering between initialisations with approximately equal spacing between features (*i.e.* neuron collapse Kothapalli (2023). In this case performance is near 100%, meaning the confusion matrix is the identity, and the superposition looses its structure. In contrast, models trained solely on CIFAR-10 converge to similar geometries because they share the same learned difficulty structure - the same pairs of classes prove challenging to distinguish, leading to consistent superposition patterns.

#### D.2.1 RESNET-18

To address concerns on architectural generalisation, we conduct the same experiments using a ResNet-18 (He et al., 2016) architecture as the base model as opposed to a ViT. We achieve a slightly higher 92% clean accuracy (compared to 89% for ViT). We observe the robust accuracy falls slightly faster for ResNet-18 than the ViT at the same epsilon values. Nevertheless, we observe similar declining trends in normalised robust accuracy and increasing transferability as bottleneck increases.

## E MODULAR ADDITION

Sec. 5 introduces a modular addition task to investigate adversarial vulnerability in a scenario where features are orthogonal. This section provides a detailed experimental setup for this task, additional results and further discussion on how we view this sort of analysis to be useful.

### E.1 LEARNED ALGORITHMIC REPRESENTATION IN MODULAR ADDITION

As detailed in Nanda et al. (2023), the network discovers and implements a trigonometric algorithm to solve this task: the encoder projects inputs onto non basis-aligned linear representations encoding

Table 8: Attack transferability (%) for $\ell_\infty$-norm PGD attacks on CIFAR-10 ViT models. Transferability is shown from a model trained with a specific 'Source Seed' (e.g., Seed 10) to three different target models, each trained with one of the seeds listed in the sub-header (e.g., 'vs. Seeds 20/30/40'). The three slash-separated values in each cell correspond to the transferability to these three target seeds, respectively. All models within a row share the same bottleneck dimension, $m$. The 'Mean ± Std' column averages transferability across all 12 source-target seed pairings for each $(\epsilon, m)$ configuration. This supports the claim in Section 4 that higher superposition can lead to more consistent latent geometries and thus higher transferability.

| $\epsilon$ | $m$ | Seed 10 vs. Seeds 20/30/40 | Seed 20 vs. Seeds 10/30/40 | Seed 30 vs. Seeds 10/20/40 | Seed 40 vs. Seeds 10/20/30 | Mean ± Std |
|---|---|---|---|---|---|---|
| | 2 | 77.8/66.7/44.4 | 25.0/62.5/25.0 | 72.7/72.7/72.7 | 58.3/75.0/58.3 | 59.3 ± 17.7 |
| 0.001 | 3 | 57.1/28.6/28.6 | 14.3/28.6/28.6 | 63.6/72.7/72.7 | 42.9/42.9/42.9 | 43.6 ± 18.4 |
| | 5 | 70.0/50.0/50.0 | 57.1/71.4/57.1 | 50.0/25.0/25.0 | 16.7/50.0/33.3 | 46.3 ± 16.9 |
| | 10 | 55.6/55.6/66.7 | 12.5/37.5/25.0 | 16.7/33.3/16.7 | 28.6/71.4/28.6 | 37.3 ± 19.3 |
| | 2 | 60.3/52.6/48.7 | 63.8/42.0/60.9 | 53.8/40.9/52.7 | 51.2/55.8/46.5 | 52.4 ± 6.9 |
| 0.01 | 3 | 42.9/46.4/50.0 | 46.4/42.9/49.1 | 44.2/38.9/44.2 | 42.9/45.1/41.8 | 44.6 ± 3.0 |
| | 5 | 43.3/41.1/43.3 | 39.2/30.4/41.8 | 46.0/40.2/47.1 | 41.2/35.3/32.9 | 40.2 ± 4.8 |
| | 10 | 42.5/46.0/44.8 | 38.0/39.4/46.5 | 43.5/47.8/47.8 | 32.6/40.0/35.8 | 42.1 ± 4.7 |
| | 2 | 35.0/36.4/37.8 | 38.8/36.0/46.3 | 37.4/31.3/41.7 | 42.0/42.0/40.7 | 38.8 ± 3.8 |
| 0.05 | 3 | 29.9/29.9/37.1 | 30.4/30.1/35.6 | 30.2/30.2/35.1 | 30.1/31.9/34.4 | 32.1 ± 2.6 |
| | 5 | 26.8/25.3/29.2 | 25.2/21.5/26.4 | 28.0/28.0/26.5 | 26.0/26.6/21.9 | 25.9 ± 2.2 |
| | 10 | 27.1/24.9/30.4 | 21.6/23.0/28.4 | 25.3/28.1/30.6 | 21.0/24.9/20.1 | 25.4 ± 3.4 |
| | 2 | 34.2/35.6/38.2 | 36.7/34.9/44.5 | 37.9/30.0/40.8 | 41.2/42.0/40.3 | 38.0 ± 3.8 |
| 0.1 | 3 | 29.6/28.9/35.2 | 28.6/29.2/34.8 | 29.1/28.2/34.0 | 28.8/31.1/32.8 | 30.9 ± 2.5 |
| | 5 | 25.4/23.3/27.0 | 23.8/19.6/24.6 | 25.9/26.7/25.4 | 24.6/24.3/20.7 | 24.3 ± 2.1 |
| | 10 | 24.8/22.6/29.7 | 20.1/21.6/26.1 | 22.5/25.5/28.0 | 19.5/23.3/18.7 | 23.5 ± 3.3 |

$\sin(\omega_k a)$ and $\cos(\omega_k a)$ at key frequencies $\omega_k = 2\pi k/P$. These features are represented orthogonally, *i.e.* not in superposition, and the sinusoidal values can be read off via linear probes. The subsequent MLP then applies sum-angle identities to compute $\cos((a+b)\,\omega_k)$ for various $k$, which constructively interfere at the logits to select the correct class. AExs are generated by relaxing the one-hot constraint and perturbing input $\mathbf{x}_j \in \mathbb{R}^P$ (encoding integer $j$) to $\mathbf{x}' = \mathbf{x} + \boldsymbol{\delta}$ via $\ell_2$ and $\ell_\infty$-norm PGD. For simplicity, only $a$ is attacked while $b$ is fixed.

As explained in Nanda et al. (2023), the network learns a structured trigonometric algorithm. $W_e$ encodes integers' sine and cosine values for key frequencies. These features are linearly decodable sine/cosine values. Scaling the corresponding feature vector alters the encoded sinusoidal value itself, not its activation intensity. Integer embeddings $\mathbf{W}_e^j$ project onto a Fourier-like basis, creating a circular/periodic structure in the embedding space. Unlike arrangements from data correlations, this circularity arises from the sine/cosine encoding of the (non-correlated) input integers.

Our model learns to utilise approximately seven key frequencies ($\omega_k$). Each embedding $W_e^j$ for an input integer $x_j$ predominantly encodes its fourteen corresponding trigonometric features: $\cos(\omega_i x_j)$ and $\sin(\omega_i x_j)$ for $i \in \{1, 2, ..., 7\}$. The concatenated 200-dim. vector thus provides these $\approx 28$ key trigonometric features (fourteen for $a$ and fourteen for $b$) to the MLP. The MLP leverages these, via sum-angle identities (e.g., $\cos(A + B) = \cos A \cos B - \sin A \sin B$), to calculate $(x_1 + x_2)$ (mod $P$). The algorithmic brittleness explored in Section 5 stems from the adversarial perturbation of these key learned trigonometric features.

Table 9: Attack transferability (%) for $\ell_2$-norm PGD attacks on CIFAR-10 ViT models. The table format, detailing source model to target model transferability, mirrors that of Table 8; please see its caption for a full explanation. These $\ell_2$ results further support the claim in Sec. 4 that higher superposition leads to increased attack transferability.

| $\epsilon$ | $m$ | Seed 10 vs. Seeds 20/30/40 | Seed 20 vs. Seeds 10/30/40 | Seed 30 vs. Seeds 10/20/40 | Seed 40 vs. Seeds 10/20/30 | Mean $\pm$ Std |
|---|---|---|---|---|---|---|
| 0.1 | 2 | 70.6/64.7/41.2 | 60.0/50.0/60.0 | 64.3/60.7/57.1 | 48.1/63.0/51.9 | 57.6 $\pm$ 8.0 |
| | 3 | 39.1/39.1/47.8 | 50.0/60.7/50.0 | 48.1/55.6/51.9 | 48.0/28.0/40.0 | 46.5 $\pm$ 8.4 |
| | 5 | 43.5/34.8/43.5 | 31.6/31.6/42.1 | 42.9/35.7/21.4 | 30.4/39.1/34.8 | 35.9 $\pm$ 6.4 |
| | 10 | 45.5/59.1/45.5 | 19.0/28.6/28.6 | 37.5/43.8/43.8 | 23.8/52.4/38.1 | 38.8 $\pm$ 11.4 |
| 0.5 | 2 | 59.6/52.1/48.9 | 60.8/35.4/60.8 | 54.1/39.4/51.4 | 43.8/49.5/46.7 | 50.2 $\pm$ 7.7 |
| | 3 | 38.7/40.6/48.1 | 36.1/36.8/41.7 | 37.4/36.5/43.5 | 34.4/34.4/36.7 | 38.7 $\pm$ 3.9 |
| | 5 | 38.4/32.0/37.6 | 29.4/25.7/32.1 | 39.1/33.9/38.3 | 31.8/29.9/32.7 | 33.4 $\pm$ 4.0 |
| | 10 | 38.3/37.4/37.4 | 26.5/28.4/31.4 | 32.4/35.3/35.3 | 23.4/28.2/25.0 | 31.6 $\pm$ 5.0 |
| 1.0 | 2 | 50.8/43.7/44.4 | 51.7/36.4/50.0 | 46.1/32.9/44.7 | 42.7/46.7/45.3 | 44.6 $\pm$ 5.3 |
| | 3 | 35.7/36.9/41.7 | 31.8/36.4/42.6 | 33.7/32.0/37.3 | 34.9/32.0/35.5 | 35.9 $\pm$ 3.3 |
| | 5 | 32.1/28.9/33.2 | 30.9/23.0/30.9 | 35.9/31.5/33.7 | 27.7/28.2/27.1 | 30.3 $\pm$ 3.3 |
| | 10 | 32.3/32.3/38.5 | 23.7/27.7/29.9 | 29.1/32.0/34.3 | 22.2/25.6/21.1 | 29.1 $\pm$ 5.0 |
| 2.0 | 2 | 47.0/41.6/44.3 | 48.2/36.2/48.2 | 43.9/33.5/42.1 | 43.4/43.4/42.8 | 42.9 $\pm$ 4.2 |
| | 3 | 31.9/37.7/40.3 | 31.1/35.6/40.6 | 32.6/31.1/36.8 | 35.4/33.3/34.3 | 35.1 $\pm$ 3.2 |
| | 5 | 29.9/25.4/32.1 | 26.3/22.1/26.7 | 32.3/31.8/29.5 | 27.9/27.9/23.6 | 28.0 $\pm$ 3.2 |
| | 10 | 30.8/30.4/35.7 | 22.5/24.9/28.6 | 24.8/29.7/32.7 | 20.4/26.1/19.0 | 27.1 $\pm$ 4.9 |
| 5.0 | 2 | 44.7/46.7/40.8 | 45.3/37.3/48.7 | 42.7/33.9/41.5 | 42.3/42.3/40.6 | 42.2 $\pm$ 3.8 |
| | 3 | 32.3/35.8/40.8 | 31.3/32.6/39.9 | 31.0/30.0/33.8 | 35.0/33.0/33.0 | 34.1 $\pm$ 3.2 |
| | 5 | 30.6/24.1/28.6 | 28.3/20.9/24.8 | 31.7/31.3/27.4 | 28.1/29.0/23.2 | 27.3 $\pm$ 3.3 |
| | 10 | 29.8/27.4/32.7 | 19.7/24.1/28.9 | 24.9/29.9/31.2 | 18.8/25.8/19.2 | 26.0 $\pm$ 4.6 |

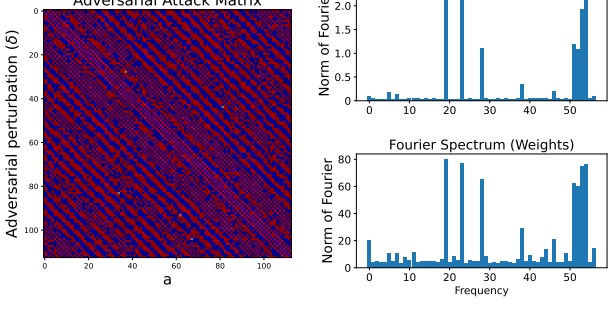

Figure 6: Left: Each column is the mean attack $\boldsymbol{\delta}$ added to $a$ to successfully attack $a + b \mod p$. Right: The norms of the Fourier components in the $\boldsymbol{\delta}$ and $W_{MLP_{out}}$.

Table 10: Comparison of minimum $\ell_2$-norm budget required to successfully attack $0 + 0 \mod p$ between different attack types.

| Attack Method | Success $\epsilon$ |
|---|---|
| PGD | 0.17 |
| Informed Attack | 0.22 |
| Single $a$ | 0.97 |
| $\ell_\infty$ Noise | 2.04 |
| Uniform $\boldsymbol{\delta}$ | 6.20 |

E.2   EXPERIMENT DETAILS FOR MODULAR ADDITION TASK

The network is trained for addition modulo 113. Inputs are integer pairs $(a, b)$, $a, b \in \{0, \ldots, 112\}$. Each integer is mapped to a 100-dimensional vector via a shared embedding $\mathbf{W}_E \in \mathbb{R}^{113 \times 100}$. Embeddings $\mathbf{W}_E^a$ and $\mathbf{W}_E^b$ are concatenated into a 200-dim. feature vector. This vector feeds a 3-hidden-layer MLP (200 ReLU neurons/layer). The output layer provides 113 logits for the

classification. The model is trained end-to-end using cross-entropy loss and AdamW (weight decay 4), achieving $100\%$ test accuracy. AExs are generated using PGD under $\ell_2$ and $\ell_\infty$ norms with varying $\epsilon$. Attacks perturb the embedding $\mathbf{W}_E^a$ (with integer $b$ fixed) by adding a perturbation $\boldsymbol{\delta}$ to cause misclassification of $(a + b) \pmod{113}$.

### E.3 Algorithmic Brittleness

The frequency-aligned perturbations cause misclassification by effectively altering the sinusoidal values read off from the encoder features. We use the hypothesised neuron-aligned case to give intuition: $\mathbf{W}_e$ would contain a row corresponding to the sine/cosine for each $\omega_k$, whose element at column $\mathbf{w}_e^a$ contains the value $\sin(\omega_k a)$ or $\cos(\omega_k a)$. For the input $a$, the one-hot $\mathbf{x}_a$ acts to select $\mathbf{w}_e^a$, giving $\mathbf{h}_a = \mathbf{w}_e^a \mathbf{x}_a$. The perturbation $\boldsymbol{\delta}$ transforms one-hot $\mathbf{x}_a$ into a dense vector $\mathbf{x}_a'$. Now $\mathbf{h}_a' = \mathbf{W}_e \mathbf{x}_a' = (1 + \epsilon_a) \cdot \mathbf{h}_a + \sum_{k \neq a} \epsilon_k \cdot \mathbf{w}_e^k$. The introduced contributions from the $k \neq a$ sinusoidal terms corrupts the sinusoidal values, thereby disrupting the MLP's subsequent computations and leading to misclassification. The model's algorithm uses the constructive interference of each $\omega_k$ at the logits, and as such disrupting a single $\omega_k$ is insufficient as the other components overwhelmingly support the correct answer. Thus, effective attacks must find $\boldsymbol{\delta}$ that maximally alters multiple frequency components.

### E.4 Analytic Construction of Attacks and Baseline Comparisons

The main paper describes how understanding the learned trigonometric algorithm in the modular addition task allows for the analytic construction of adversarial attacks. Sec. 5 of the main paper benchmarks this 'informed attack' against PGD and naive baselines. This section provides further details on these baseline attack methods. The baselines are:

- **PGD Attack:** Standard PGD used to find the adversary.
- *Informed Attack:* The analytically constructed perturbation, from combinations of cosines.
- *Single $a$:* This involved finding the minimal attack to a single input to cause misclassification.
- $l_\infty$ *Noise:* This baseline involved adding random noise $\delta_{noise}$ to the input a. Each component of $\delta_{noise}$ was sampled independently and uniformly from $[-\epsilon, \epsilon]$. The $\epsilon$ that successfully induced a misclassification is reported.
- *Uniform $\boldsymbol{\delta}$:* The perturbation vector $\boldsymbol{\delta}$ is constrained to have all components equal, i.e., $\boldsymbol{\delta}_j = c \cdot \mathbf{1}$, where $c$ is a scalar. The magnitude of $c$ is reported such that adding $\boldsymbol{\delta}$ to the input causes a misclassification.

### E.5 Details on Certified Robust Training

To investigate the impact of defences on the identified vulnerability, we performed certified robust training. This aimed to make the network provably robust against $l_\infty$-norm perturbations. The certified training method employed was an abstraction-based technique (Wong & Kolter, 2018; Raghunathan et al., 2018; Palma et al., 2022), namely RSIP-IBP (Zhang et al., 2020; Henriksen & Lomuscio, 2023). RSIP-IBP utilises Interval Bound Propagation (IBP) (Katz et al., 2017; Ehlers, 2017) and Reversed Symbolic Interval Propagation (RSIP) to compute reachable output sets for network layers, thereby enabling formal certification of robustness within a given perturbation budget. The training loss function incorporated both a standard loss component and an RSIP loss component. Certified training was conducted for 1000 epochs, with the initial 200 epochs serving as a warm-up phase. The learning rate was $1 \times 10^{-3}$ and weight decay $1 \times 10^{-4}$. The same train/test data splits were used as for the standard model training. The results of certified training across different training perturbation budgets ($\epsilon_{train}$) and evaluated against various attack perturbation budgets ($\epsilon_{attack}$) are summarised in Table 11.

As discussed in Sec. 5, when these robustly trained model were subsequently attacked using PGD, the nature of successful adversarial perturbations remained consistent: they continued to exploit the same key frequency components inherent to the model's learned algorithm. The primary effect of the certified training was an increase in the perturbation magnitude required to achieve misclassification for those epsilons against which the model was successfully certified, rather than a fundamental change in the attack vector's alignment with the model's algorithmic sensitivities.

Table 11: Verified robust accuracy (%) for modular addition models under $\ell_\infty$-norm perturbations. The table compares a standard (pre-robustification) model with models certifiably trained using RSIP-IBP with different training perturbation budgets ($\epsilon_{\text{train}}$). Accuracies are evaluated against various attack budgets ($\epsilon_{\text{attack}}$). These results support the discussion in Section 5.1 of the main paper, indicating that certified training improved robustness to specified $\epsilon_{\text{train}}$ but attacks still succeeded by exploiting the same algorithmic sensitivities at higher $\epsilon_{\text{attack}}$.

| Attack Epsilon ($\epsilon_{attack}$) | Train Epsilon ($\epsilon_{train}$) | Standard Acc. (Pre-Rob., %) | Robust Acc. (Pre-Rob., %) | Standard Acc. (Robust, %) | Robust Acc. (Robust, %) |
|---|---|---|---|---|---|
| $5.0 \times 10^{-4}$ | $10^{-4}$ $10^{-3}$ | 100.0 | 100.0 | 100.0 | 100.0 100.0 |
| $1.0 \times 10^{-3}$ | $10^{-4}$ $10^{-3}$ | 100.0 | 76.0 | 100.0 | 93.8 100.0 |
| $2.0 \times 10^{-3}$ | $10^{-4}$ $10^{-3}$ | 100.0 | 0.0 | 100.0 | 0.0 95.7 |
| $5.0 \times 10^{-3}$ | $10^{-4}$ $10^{-3}$ | 100.0 | 0.0 | 100.0 | 0.0 85.3 |

### E.6 ON A POTENTIAL RECONCILIATION WITH SUPERPOSITION VULNERABILITY

Although the vulnerability explored here arises in the absence of feature superposition, it exhibits a notable phenomenological resemblance to those driven by superposition — a shared dependence on sparsity in the input representation. This motivates our *reconciliation conjecture*: algorithmic vulnerabilities may persist in networks with orthogonal representations when sparsity assumptions are violated upstream. If earlier layers in a network produce noisy or non-sparse features, potentially due to adversarial interference, then these deviations can propagate into otherwise well-isolated downstream circuits. In such cases, circuits that operate correctly on clean inputs may nonetheless fail, as the input they receive is effectively already perturbed. This provides a plausible route by which algorithm-disruption mechanisms, like the frequency attacks studied here, could be activated even in the absence of superposition. Further, this interaction between upstream superposition and downstream algorithmic brittleness, and more broadly layer-by-layer understanding of the specific vulnerability mechanisms triggered by AExs, is underexplored. We leave a detailed investigation, grounded in empirical evidence and theoretical modelling, as a promising direction for future work.

## F INSPECTING ADVERSARIAL PERTURBATIONS VIA SAE FEATURES

The analyses of the main paper focused on scenarios with either explicitly defined class features (toy model) or class-level representations in a bottleneck (ViT experiments). However, in large, practical networks, interference will occur between a unlabeled latent features across all layers. As discussed in the future work section of the main paper, SAEs offer a promising unsupervised method to extract more linear features from NNs (Bricken et al., 2023). This appendix section presents a preliminary investigation into using SAE-extracted features to characterise how a large scale ViT responds internally when processing AExs versus clean inputs. These initial results are intended to explore the feasibility of this approach for future work aimed at understanding adversarial phenomena on a feature level within large models, rather than presenting a conclusive new set of findings. We first outline the experimental setup and validate the SAEs behavior on adversarial inputs before presenting exploratory layer-wise feature difference analyses.

### F.1 EXPERIMENT SETUP

We analyse SAE-extracted features from the vision encoder of the LAION/CLIP-ViT-B-32-DataComp.XL-s13B-b90K model, a 12-layer ViT with $\approx 70\%$ zero-shot accuracy on ImageNet-1k Deng et al. (2009). We use open-source Prisma SAEs mapping the 768-dimensional activation space to a dictionary of 49,152 features ($64\times$ expansion) (Joseph et al., 2025). We examine SAEs

trained on either mean patch token representations or CLS token representations, across all layers. We analyse features activating above a high threshold (0.1) to capture changes to the highest activating features, and a low threshold (0.001). We generate 10,000 original/successfully-attacked image pairs ($\ell_\infty$-norm, $\epsilon = 0.02$) for our SAE feature comparisons. This setup allows for an initial exploration of how adversarial perturbations might manifest at the level of SAE features.

## F.2 SAE Validity for Adversarial Inputs

Given that AExs are out-of-distribution inputs, and our aim is to use SAE (standardly trained) to understand them, we first conduct a basic validation. This subsection examines the impact on model accuracy and SAE reconstruction when SAEs are inserted at inference time for both clean and adversarial inputs. We are investigating whether the SAEs process features relevant to the classification task, even under adversarial conditions, before attempting to interpret differences in feature activations.

Table 12: Impact on ImageNet-1k classification accuracy when inserting a pre-trained SAE (one layer at a time) during inference for original, $\ell_\infty$ adversarial ($\epsilon = 0.02$), and random noisy inputs. 'Accuracy' refers to the ViT's top-1 accuracy. 'Original input + SAE' means the clean ViT activations at a given layer are passed through the SAE then its reconstruction is passed to the next layer. The slight 'de-attacking' effect (improved accuracy for 'Adv. input + SAE' vs 'Adv. input') suggests SAEs might filter some adversarial noise, but overall, the model still largely misclassifies, indicating salient adversarial changes for misclassification are processed.

| Metric | Layer 2 | Layer 5 | Layer 8 | Layer 11 |
|---|---|---|---|---|
| **Accuracy** | | | | |
| Original input | 1.00 | 1.00 | 1.00 | 1.00 |
| Original input + SAE | 0.99 | 0.99 | 0.99 | 0.86 |
| Adv. input | 0.00 | 0.00 | 0.00 | 0.00 |
| Adv. input + SAE | 0.02 | 0.01 | 0.01 | 0.11 |
| Noisy input | 0.98 | 0.98 | 0.98 | 0.98 |
| Noisy input + SAE | 0.98 | 0.98 | 0.98 | 0.85 |
| **SAE Improvement** | | | | |
| Original input | -0.01 | -0.01 | -0.01 | -0.14 |
| Adv. input | 0.02 | 0.01 | 0.01 | 0.11 |
| Noisy input | -0.00 | -0.00 | -0.00 | -0.13 |

## F.3 Layer-wise SAE Feature Differences

This subsection presents initial exploratory visualisations comparing SAE feature activations across different layers of the ViT for original, $\ell_\infty$ adversarial, and random noisy inputs. These visualisations are intended to highlight potential avenues for future, more in-depth investigation into how adversarial attacks distinctively alter sparse feature representations. We examine:

- Activation count: Histograms comparing the number of SAE features activated above a set threshold in a layer for original, adversarial, and noisy images. This explores whether attacks characteristically alter feature sparsity.
- Feature overlap: The degree of overlap (number of common activated features) between (original vs. adversarial) and (original vs. noisy) conditions. This explores if attacks activate a distinctly different set of features.
- Jaccard Similarity: The Jaccard similarity between sets of activated features, offering a normalised measure of overlap.
- Mean activation: Histograms of mean activation values for features active above a threshold. This looks for changes in the intensity of (key) features.
- Distinct Features: The count of unique features activated across an image set for each condition.

### F.4 CLS TOKEN SAE FEATURES WITH HIGH ACTIVATION THRESHOLD (0.1)

This subsection presents a preliminary analysis of SAE features derived from CLS tokens, only considering features activated above a threshold of 0.1. Figures 7 through 10 visualise and compare the histogram of activation counts, feature overlap, Jaccard similarity, and histogram of mean activations, respectively, for original, adversarial, and random noisy inputs across different ViT layers.

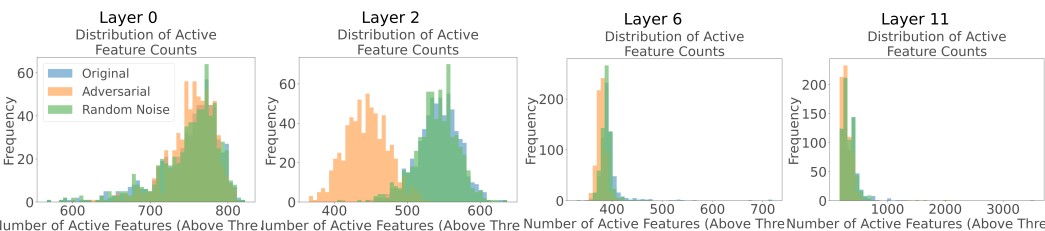

Figure 7: Histogram of activation counts for CLS token SAE features (threshold 0.1)

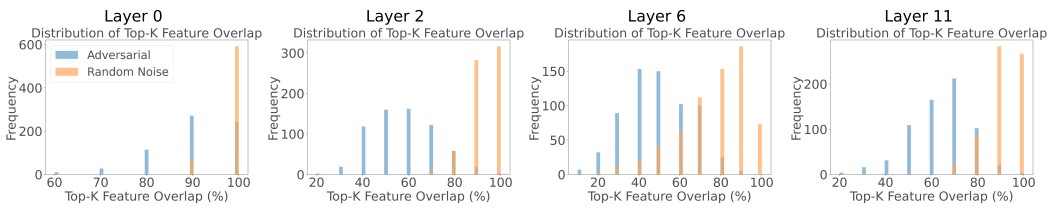

Figure 8: Feature overlap between original vs. attacked and original vs. noisy images for CLS token SAE features (threshold 0.1)

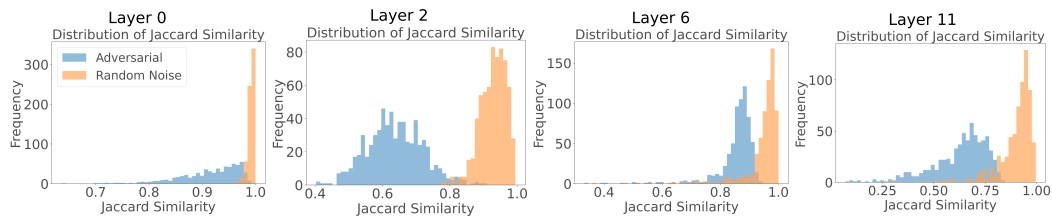

Figure 9: Comparison of Jaccard similarity for CLS token SAE features (threshold 0.1)

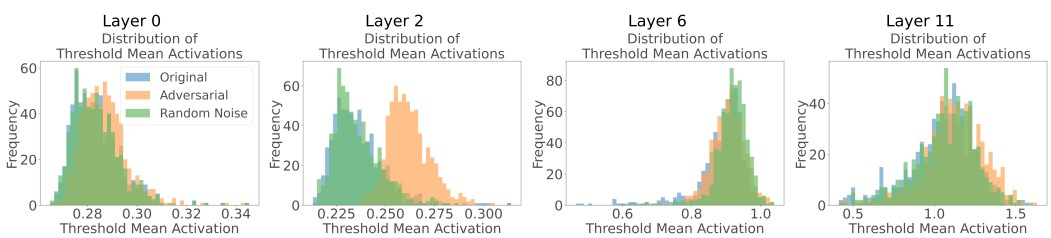

Figure 10: Histogram of mean thresholds for CLS token SAE features (threshold 0.1).

### F.5 CLS TOKEN SAE FEATURES WITH LOW ACTIVATION THRESHOLD (0.001)

Here, we examine SAE features from CLS tokens with a lower activation threshold of 0.001, to not just capture the most intensely activated features. The subsequent figures illustrate the histogram of activation counts (Figure 11), feature overlap (Figure 12), Jaccard similarity (Figure 13), and histogram of mean activations (Figure 14) across layers for original, adversarial, and noisy inputs.

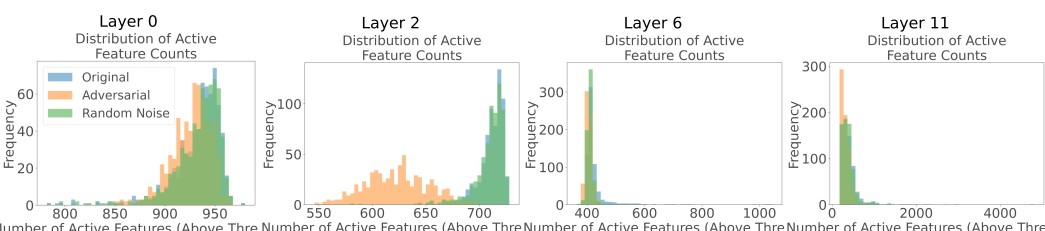

Figure 11: Histogram of activation counts for CLS token SAE features (threshold 0.001), with noise.

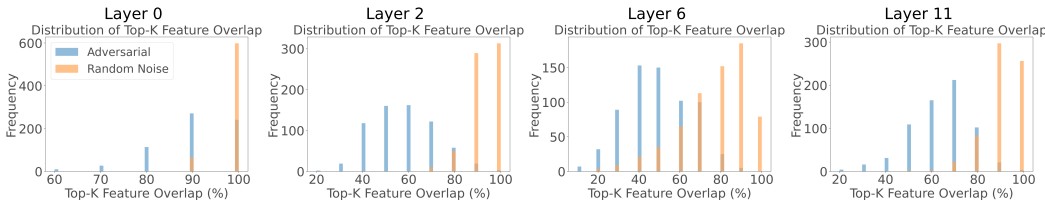

Figure 12: Feature overlap between original vs. attacked and original vs. noisy images for CLS token SAE features (threshold 0.001)

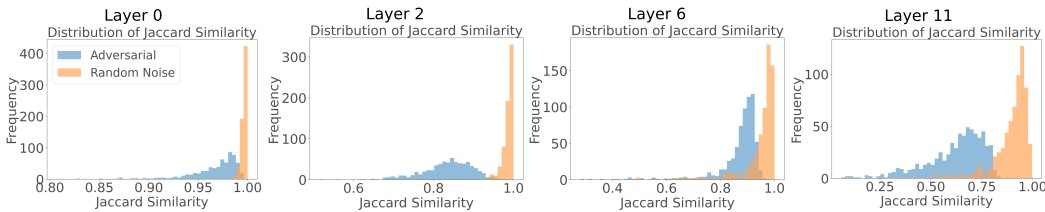

Figure 13: Comparison of Jaccard similarity for CLS token SAE features (threshold 0.001).

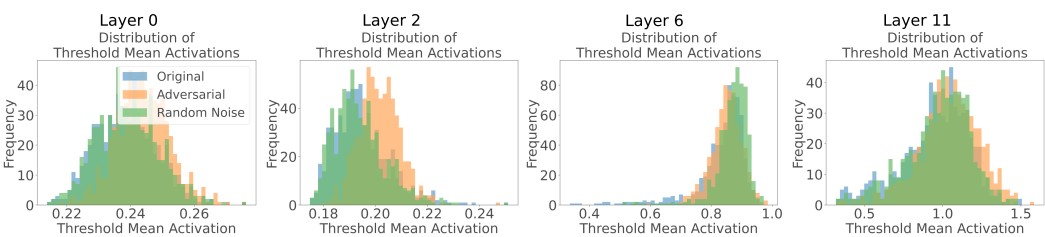

Figure 14: Histogram of mean activations for CLS token SAE features (threshold 0.001), with noise.

## F.6    PATCH TOKEN SAE FEATURES WITH HIGH ACTIVATION THRESHOLD (0.1)

This subsection shifts the focus to SAE features derived from the mean of patch tokens, again using a high activation threshold of 0.1. We present comparisons of activation count histograms (Figure 15), distinct feature count histograms (Figure 16), feature overlap (Figure 17), Jaccard similarity (Figure 18), and mean activation histograms (Figure 19) for original, adversarial, and noisy inputs.

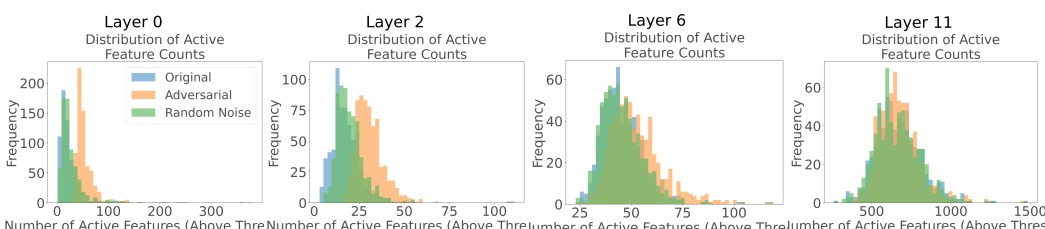

Figure 15: Histogram of activation counts for patch token SAE features (threshold 0.1), with noise.

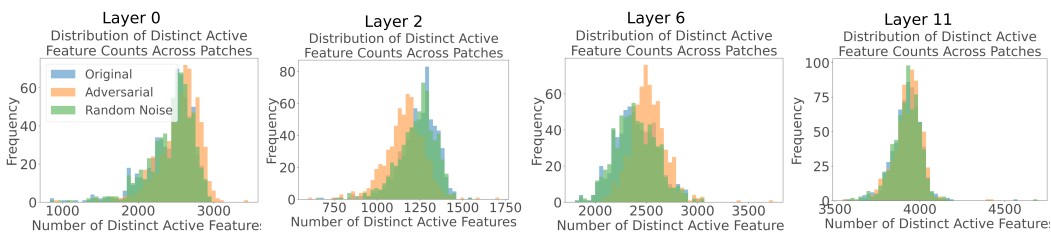

Figure 16: Histogram of distinct patch features for patch token SAE features (threshold 0.1), with noise.

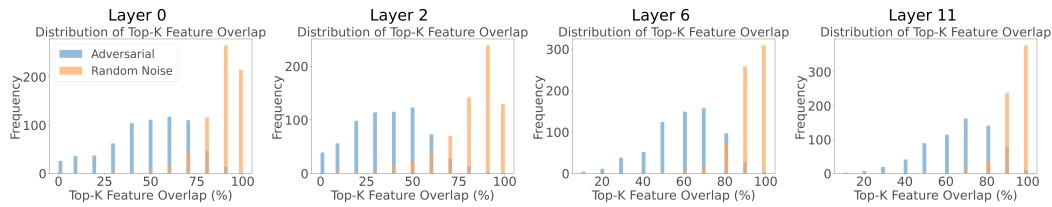

Figure 17: Feature overlap between original vs. attacked and original vs. noisy images for patch token SAE features (threshold 0.1).

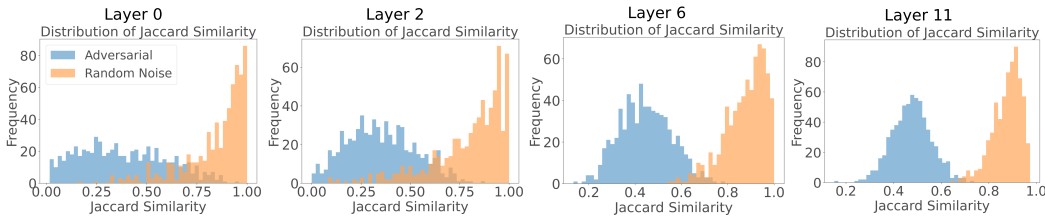

Figure 18: Comparison of Jaccard similarity for patch token SAE features (threshold 0.1).

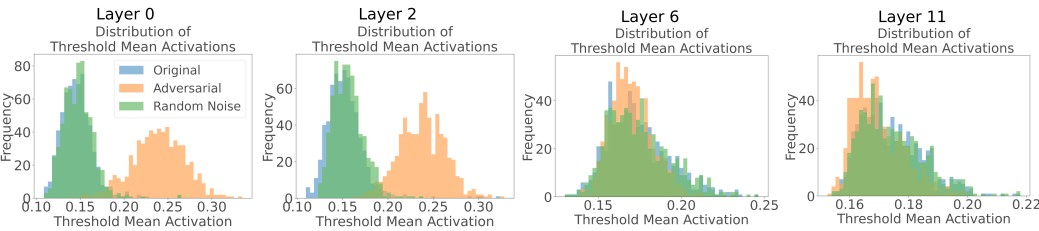

Figure 19: Histogram of mean thresholds for patch token SAE features (threshold 0.1), with noise.

## F.7 PATCH TOKEN SAE FEATURES WITH LOW ACTIVATION THRESHOLD (0.001)

Finally, this subsection explores SAE features from the patch tokens means using a low activation threshold of 0.001. The figures show histograms of activation counts (Figure 20), distinct feature counts (Figure 21), feature overlap (Figure 22), Jaccard similarity (Figure 23), and mean activation histograms (Figure 24) for original, adversarial, and noisy inputs.

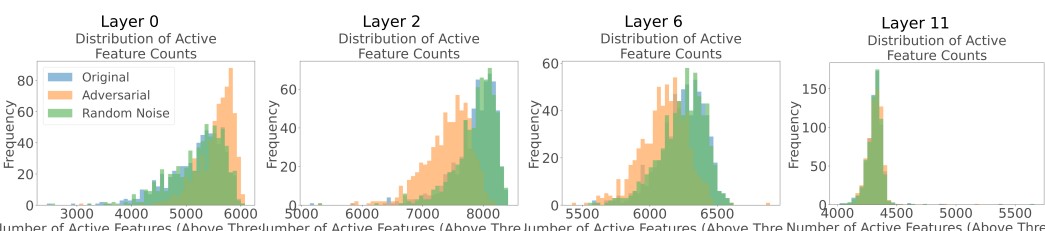

Figure 20: Histogram of activation counts for patch token SAE features (threshold 0.001), with noise.

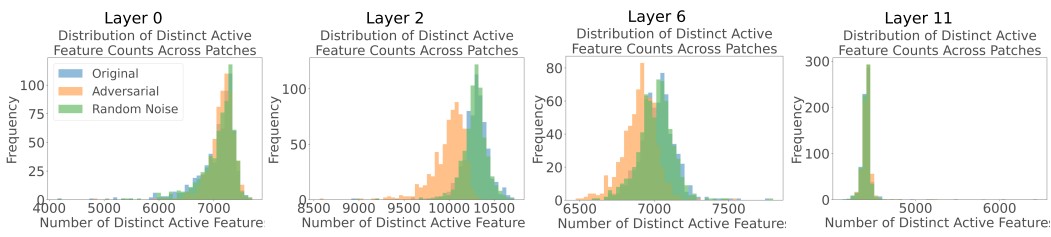

Figure 21: Histogram of distinct patch features for patch token SAE features (threshold 0.001), with noise.

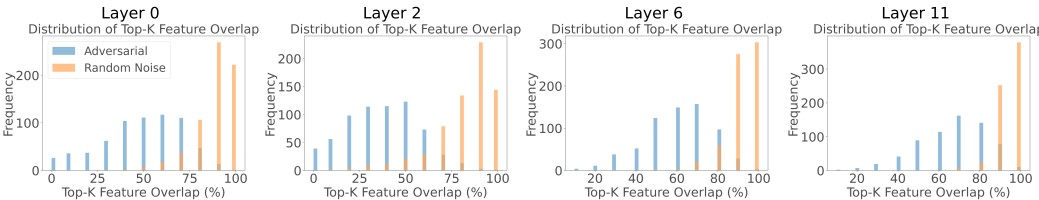

Figure 22: Feature overlap between original vs. attacked and original vs. noisy images for patch token SAE features (threshold 0.001).

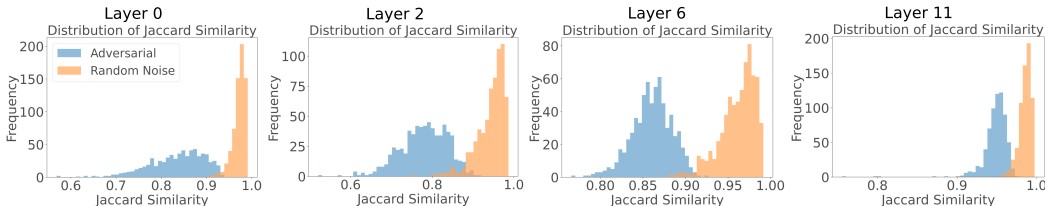

Figure 23: Comparison of Jaccard similarity for patch token SAE features (threshold 0.001).

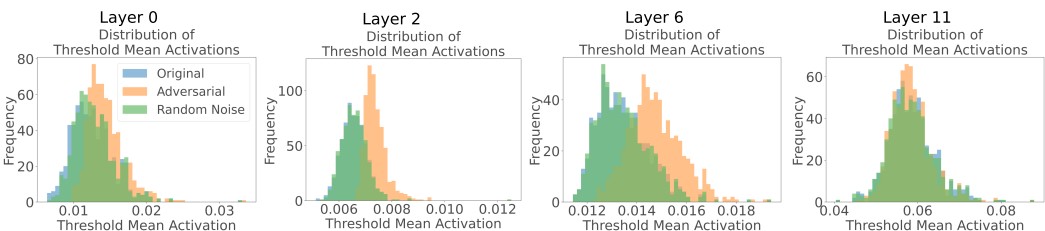

Figure 24: Histogram of mean activations for patch token SAE features (threshold 0.001), with noise.

# G    ACRONYMS

**AEx**  adversarial example

**CE**  cross entropy

**LRH**  linear representation hypothesis

**MLP**  Multilayer Perceptron

**MSE**  mean squared error

**NN**  neural network

**PGD**  projected gradient descent

**SAE**  sparse autoencoder

**ViT**  vision transformer

