# OpenReview forum: "Adversarial Attacks Leverage Interference Between Features in Superposition"
_ICLR.cc/2026/Conference — Submitted to ICLR 2026_

### Official Review · Reviewer_b9T9 · 2025-10-31

**Soundness:** 2
**Presentation:** 2
**Contribution:** 2
**Rating:** 0
**Confidence:** 3

**Summary:**

The authors explore the existence of adversarial examples in neural networks.
They propose that superposition in feature space is created through dimensionality reduction, so that several input features change the same intermediate node in similar ways.
They experiment with artificial random data vectors and a two-layer linear network, as well as with CIFAR-10 and ViT as feature extractor, for which they train another two-layer linear network.
They show that changes to the input features lead to a predictable change in the output.

**Strengths:**

-

**Weaknesses:**

My main issue with this paper is that it solely investigates adversarial samples in linear models.
It is absolutely unclear if or how the conclusions drawn this paper would transfer to realistic models which are highly non-linear.


1. It is quite difficult to follow the argumentation. The terms "input" and "feature" are used in many different ways, without properly introducing them. It seems that sometimes, the term represents data elements, and sometimes network parameters or dimensions in feature spaces:

   a) Line 53 mentions some "input features" without providing any intuition what this would be, or what they would be input to.

   b) In line 104, it is mentioned that individual neurons contribute to multiple "features", but without understanding what a "feature" is, such a claim is meaningless.

   c) Line 110 uses the term number of "latent features" but it is unclear what this refers to.

   d) Line 132 discusses "input" shape, but it is not clear whether the authors refer to "input features" or to images.

   e) Line 141 discusses "input features" (which represent single data points) into class-associated groups -- this makes no sense since each input comes only from one class and, thus, this grouping cannot be done.


2. The paper is not clear in which kinds of concepts they want to investigate:

   a) In line 81, the authors define a set of "semantically meaningful [...] concepts", but there is no intuition given that a deep network learns such concepts.

   b) In line 137, the concepts are defines to be "class concepts", which are typically given as the final fully-connected layer, but they do not correspond to any semantics.


3. There are some conceptual issues within the experiments:

   a) In line 143, the authors just sum up features from groups to determine the assigned class, which would require that feature magnitudes relate to class contribution, and that magnitudes are comparable between classes. There is no reason to believe that this would be true for features extracted from real data.

   b) In line 145, input features are sampled from a uniform distribution, which indicates that elements in there are not correlated. This is clearly not what happens when you extract "input features" from real data. The expressiveness of this experiment is, therefore, low.

   c) In line 148, the authors use cross-entropy loss to train their two-layer network. It is unclear what the task is that they train for, what are the categories that they want to compare to, or how can they use MSE loss (line 151) for the same task as CE.

   d) Figure 1 shows an "adversarial example" that has very different data restrictions as their original data. In figure 1(b), input values can be negative, while original inputs are only positive. Hence, detecting this adversarial example would be simple, making it not to be adversarial (by the definition of adversariality, line 454).

   e) Figure 1(c) shows that the adversarial sample moves **linearly** in latent feature space. For their two-layer network, such a linearity assumption is useful since the model is linear. However, there is clear evidence [Rozsa2018, Figure 5.2] that a real adversarial sample (where we perturb the input and not the features) do not move linearly in latent feature space. Particularly, Propositions 1 (line 312) and 2 (line 327) clearly do not hold for deeper, non-linear networks.

   f) In line 269, the authors mention "geometric arrangements" of models, but it is unclear what "geometry" means here. Do they mean network topologies?


4. The evaluation of the proposed theory is not convincing:

   a) The authors only use a single adversarial image generation technique (PGD) which is well-known to induce perturbations across the entire input. More localized techniques, such as Carlini&Wagner need to be explored. Also, PGD has several parameters, but the authors do not provide any details.

   b) The authors only experiment with a single image dataset, CIFAR-10, which is known to be very noisy.

   c) It is unclear why the authors performed a two-stage training process for CIFAR-10. When training the network including the bottleneck directly leads to better-separated feature spaces, so figure 4(a) would rather look like figure 2(a).

   d) Also, it is not clear what the authors have changed via PGD: the original input images (non-linear operation), or the features extracted by the network (a linear operation). Line 390 uses variable $\mathbf x$, which was used for input features before.


5. While this paper provides a theoretical analysis, where a clear mathematical formulation would be required, there are several issues within the math:

   a) In line 100, the authors use the symbol $\Theta(de^2)$ without defining any of the parameters in there.

   b) The nomenclature changes between definitions. While Definition 3 uses $f(x)$ to represent the classifier, Definition 4 uses $g_w(x)$. The authors should be consistent.

   c) Line 402 uses "one-hot integers" $(a,b)$ and $P$, but it is entirely unclear what this would be.


[Rozsa2018] Andras Rozsa: "Towards Robust Deep Neural Networks", PhD thesis, University of Colorado Colorado Springs, 2018

**Questions:**

That linearity constraints hold in linear networks is obvious, and we do not need this analysis. The most important question would be whether such assumptions also hold for deep, non-linear networks, but the authors do not investigate this at all.

---

> ### Author Response · Authors · 2025-11-29
> **Comment by Authors (Part 1/3)**
>
> We thank the reviewer for their time and feedback. However, we are concerned that the review does not acknowledge any strengths of the work and appears to overlook several key contributions and factual points. We address all the reviewer’s comments in detail below and clarify where some statements are inaccurate or based on misunderstandings. In light of this, we respectfully ask the AC and the other reviewers to consider our clarifications carefully, as we believe the evaluation provided in this particular review does not reflect the technical content or merit of the submission.
>
> In what follows, we structure our response into three categories to facilitate clear engagement:
>
> 1. New evidence addressing the primary concerns about linearity and experimental constraints.
> 2. Clarifications of our methodology and definitions.
> 3. Pointers to where specific raised issues were already addressed in the original submission.
>
> ## **New experiments**
>
> ### **1. Non-linearity concerns.**
> The reviewer’s main concern is that "this paper solely investigates adversarial samples in linear models". **We like to point out that this is not true.** Our toy model analysis (Section 3) includes setups with a softmax/ReLU, and that our Section 4 experiments are on standard ViTs with non-linear activations, through which attacks must pass (though superposition is shown in a linear bottleneck).
>
> However, we agree that the Section 3 results are mainly established in a limited-depth setting. To demonstrate that our conclusions transfer to deeper, highly non-linear architectures, **we now introduce a new experiment using a standard ViT architecture on a distributed version of the argmax task**.
>
> **Setup.** We use a transformer with $K$ classes distributed across $N$ tokens. Inputs are compressed through a residual stream of dimension $m < K$ to force superposition. The model must aggregate information across tokens via attention to identify the class with the highest total magnitude. This architecture includes all standard non-linear ViT components (CLS token, attention heads, residual stream).
>
> **Results.** Consistent with our toy model findings, we observe that adversarial examples systematically exploit superposition. We measure this via the cosine similarity between PGD-discovered perturbation directions and the optimal perturbations derived from the superposition
> geometry in each of the tokens. The alignment remains high (cosine sim $>0.77$), confirming that the vulnerability arises from feature interference.
>
> ### **2. Negative values in attacks**
>
> The reviewer notes that "input values can be negative, while original inputs are only positive". We chose an unconstrained input domain for the toy model attacks to provide the clearest geometric intuition. However, as pointed out, these negative perturbations fall outside the training distribution, and to demonstrate this is not a validity issue, **we re-ran the experiment from Table 1 with the constraint that perturbed inputs remain [0, 1]**. While the absolute robustness increases (as the attack is more constrained), the mechanism remains the same: successful attacks still align with the theoretically directions predicted by superposition geometry.
>
> We also note this does not invalidate our definition of an adversarial example, which is that (a) the model’s prediction under perturbation changes (e.g. from class A to B) and (b) the true class label remains unchanged (e.g. the ground truth is still class A).

---

> > ### Author Response · Authors · 2025-11-29
> > **Comment by Authors (Part 2/3)**
> >
> > ## **Clarifications**
> >
> > We now provide clarifications to feedback, following the same numbering system as used by the reviewer.
> >
> > **1:** *[The terms “input” and “feature” are used in many different ways, without properly introducing them.]*
> >
> > The input is the input to a model, and we distinguish between "input features" (the semantic properties of data, e.g., 'has ears') and "latent features" (linear directions in activation space representing those variables). Input and latent feature are formally defined on lines 73 and 80-89, respectively. Input features are defined for our task in 142.
> >
> > **1 (d):** *[Line 132 discusses "input" shape, but it is not clear whether the authors refer to "input features" or to images]*
> >
> > The correlations are introduced between the elements of the input vector to this task, i.e. the ‘uncorrelated’, ‘correlated pairs, and ‘global’ conditions defined in the subsection entitled “Do input correlations determine latent feature geometry?” (l.230)
> >
> > **3 (a)** *[In line 143, the authors just sum up features from groups to determine the assigned class, which would require that feature magnitudes relate to class contribution, and that magnitudes are comparable between classes. There is no reason to believe that this would be true for features extracted from real data.]*
> >
> > While the "largest sum" rule is a simplification, it aligns with the Linear Representation Hypothesis, where activation magnitude signifies feature presence. Growing literature [1, 2, 3] confirms networks rely on these linear magnitudes for computation. Our model captures the intuition that attacks function by manipulating these activation strengths—specifically, amplifying features of incorrect classes to outweigh evidence for the true class.
> >
> > **4 (c)** *[It is unclear why the authors performed a two-stage training process for CIFAR-10.]*
> >
> > We employ two-stage training to isolate the effect of superposition. We want to remove changes in earlier layers as a source of increased vulnerability.

---

> > > ### Author Response · Authors · 2025-11-29
> > > **Comment by Authors (Part 3/3)**
> > >
> > > ## **Pointing to manuscript**
> > >
> > > **1 (a)** *[No intuition is provided for "input features" or what they would be input to when mentioned on l.53.]*
> > >
> > > l.54: We describe input features as “fundamental abstractions of data” and on l.81 give examples of input features that may be linearly represented in latent space, “presence of shape,” or “indoor vs. outdoor”.
> > >
> > > l.143: In our toy model simplification, we give the intuition that each input dimension “represents evidence for class j, with the p features capturing different class attributes” (defining those terms on l.141-142).
> > >
> > > **1 (b) and (c):**  *[In l.104, it is mentioned that individual neurons contribute to multiple "features", but without understanding what a "feature" is, such a claim is meaningless.] and [l.110 uses the term number of "latent features" but it is unclear what this refers to]*
> > >
> > > l.81-89 we formally define latent features. Individuals neurons contributing to multiple features (polysemanticity) is introduced on l.103-105, literature provided [e.g. 4] on l.103, and explained via superposition on l.108-112.
> > >
> > > **1 (c):** *[Line 141 discusses "input features" (which represent single data points) into class-associated groups -- this makes no sense since each input comes only from one class and, thus, this grouping cannot be done.]*
> > >
> > > l.142 states that each class can contain p input features. Some experiments use p=1, in which case each class contains only one element of the input vector **x**.
> > >
> > > **2 (a):** *[In line 81, the authors define a set of "semantically meaningful [...] concepts", but there is no intuition given that a deep network learns such concepts.]*
> > >
> > > l.77: We explicitly point to research establishing this; l.78 intuition is given as to why linear representations occur; l.445-451 in our related work we point to numerous works that establish this.
> > >
> > > **3 (b):** *[In line 145, input features are sampled from a uniform distribution, which indicates that elements in there are not correlated. This is clearly not what happens when you extract "input features" from real data…]*
> > >
> > > l. 230-238: The reviewer describes the ‘uncorrelated’, i.i.d condition in the paper (replicating [5]). The same paragraph then establishes the ‘correlated pairs’ and ‘global correlations’ conditions to introduce more realistic correlations.
> > >
> > > **3 (c):** *[In l.148, the authors use cross-entropy loss to train their two-layer network. It is unclear what the task is…]*
> > >
> > > l.142 states: “The task identifies which group has the largest sum: y = …”
> > >
> > > We minimise MSE between the outputs and one‑hot class labels, i.e., treating classification as regression on probabilities.
> > >
> > > **3 (f):** *[In l.269, the authors mention "geometric arrangements" of models, but it is unclear what "geometry" means]*
> > >
> > > l.240: “We quantify geometric similarity by comparing the pairwise cosine similarity matrices between all feature pairs in each model, then measuring the correlation between these matrices across different random seeds. ”
> > >
> > > **4 (a):** *[PGD has several parameters, but the authors do not provide any details.]*
> > >
> > > e.g. l.1059: We provide iterations, step size, etc.
> > >
> > > **5 (a)** *[In l.100, the authors use the symbol without defining any of the parameters in there.]*
> > >
> > > l.100 We use “2^{Θ(d\epsilon^{2})}” and define ϵ and d in the same sentence, whilst theta is big-Theta notation from asymptotic analysis. Regarding 5(b), we agree that f(x) and g(x) can be made consistent.
> > >
> > > **5 (c):** [l.402 uses "one-hot integers" (a, b) and P, but it is entirely unclear what this would be.]
> > >
> > > $a$ and $b$ are one-hot vectors representing integers. l.1231-1285: We provide extended details in the app. and also provide references to the original paper.
> > >
> > > **4 (d)** *[It is not clear what the authors have changed via PGD: the original input images…]*
> > >
> > > All perturbations are on the input to the model, throughout the paper.

---

> > > > ### Author Response · Authors · 2025-11-29
> > > >
> > > > ## **Other**
> > > >
> > > > **2 (b)** *[In l.137, the concepts are defined to be "class concepts", which are typically given as the final fully-connected layer, but they do not correspond to any semantics.]*
> > > >
> > > > We employ "class concepts" to ensure ground truth access, allowing us to track how inputs correspond to superposed representations. These concepts correspond to semantics in that they are what is needed to be represented to achieve the task, and can be ablated or steered to alter predictions, demonstrating they function as units of computation
> > > >
> > > > **3 (e)** *[Fig. 1(c) shows that the adversarial sample moves **linearly** in latent feature space]* and *[…real adversarial sample (where we perturb the input and not the features) do not move linearly in latent feature space.]*
> > > >
> > > > The arrow in Fig. 1(c) shows the vector between the original and adversarial sample—by definition a straight line. This does not imply the PGD trajectory is linear. Furthermore, we note that linear bottlenecks with downstream non-linearities are common in practice (e.g., LLM embedding matrix).
> > > >
> > > > **4 (a)** *[The authors only use a single adversarial image generation technique (PGD) which is well-known to induce perturbations across the entire input.]*
> > > >
> > > > l.478. This is a good question, which we are keen to investigate, and note in the limitations.
> > > >
> > > > ## **References**
> > > >
> > > > [1] The Linear Representation Hypothesis and the Geometry of Large Language Models, Park et al., ICML, 2024.
> > > >
> > > > [2] Steering CLIP's vision transformer with sparse autoencoders, Joseph et al., 2025 , 2024
> > > >
> > > > [3] Steering Llama 2 via Contrastive Activation Addition, ACL, 2023
> > > >
> > > > [4] Polysemanticity and Capacity in Neural Networks, Scherlis et al., 2022
> > > >
> > > > [5] Toy Models of Superposition, Elhage et al., 2022

---

### Official Review · Reviewer_SnJF · 2025-11-01

**Soundness:** 2
**Presentation:** 2
**Contribution:** 2
**Rating:** 4
**Confidence:** 2

**Summary:**

This paper argues that the adversarial vulnerability of neural networks stems from efficient information encoding in NNs. The adversarial vulnerability emerges from the interaction between architectural constraints and data semantics, Superposition creates arrangements of latent representations that adversaries can exploit.

**Strengths:**

1. The paper provides a novel explanation for adversarial vulnerablility that it comes from the interference between superposed features in neural representations.
2. The superstition framework provides an explanation for adversarial attack transferability.

**Weaknesses:**

1. The main body is based on empirical observations, which are based on a synthetic data set. It makes sense that authors need to enable testable predictions about adversarial mechanisms, though.
2. With reduced superposition, does the robustness improve?

**Questions:**

1. When the features are disentangled, is there evidence that a linear classifier is less vulnerable?

---

> ### Author Response · Authors · 2025-11-29
>
> We thank the reviewer for their feedback. We believe that we address all the concerns of the reviewer below and kindly ask them to reconsider their final evaluation.
>
> ### **W1. On the use of synthetic dataset for main experiments**
>
> We appreciate this observation and want to clarify our choice to use a synthetic dataset, which enables us to:
>
> - As the reviewer notes, know the ground-truth correspondence between input features and learned representations.
> - Precisely control superposition pressure ($k/m$ ratio)
> - Make testable, quantitative predictions about attack patterns (e.g., Table 1 shows cosine similarity between predicted and actual attacks)
>
> We validate our core findings on CIFAR-10 in a ViT (Figure 4) and ResNet-18 (Appendix D.5), replicating our findings that increased superposition decreases robustness and increases transferability.
>
> We point to Appendix F which shows preliminary analysis on how  feature extraction techniques (e.g., SAEs [1]) could be used in this type of analysis on large-scale models. We believed that the reconstruction errors and confounding factors these methods introduce would obscure our argument about superposition being **sufficient** for vulnerability.
>
> ### **W2. Does reduced superposition improve robustness?**
>
> Yes, we provide extensive evidence that reducing superposition improves robustness:
>
> 1. In Section 3.1 ($m=k$ experiment), when superposition is eliminated, we find **zero successful adversarial examples** across 1000 attempts at all $\epsilon$ values tested. Any perturbation changing the model's prediction also changes the ground truth class.
> 2. In our capacity-controlled experiment (Table 5) we fix $m$ and vary $k$, which leads to robust accuracy decreases monotonically with superposition pressure $k/m$, isolating superposition's effect from model capacity.
> 3. In the orthogonal feature experiment (Figure 3) attacks avoid perturbing features represented orthogonally, focusing their budget exclusively on superposed features.
> 4. In the CIFAR-10 ViT experiments (Figure 4, Tables 6-7), as bottleneck dimension increases (reducing superposition), both normalised robust accuracy increases and attack transferability decreases.
>
> We highlight the point we make in Section 5, that superposition is **sufficient but not necessary** for vulnerability. We demonstrate "algorithmic brittleness" as a distinct mechanism where orthogonal representations can still be vulnerable when downstream computations are sensitive to precise feature values.
>
> ### **Q1. When features are disentangled, is there evidence that a linear classifier is less vulnerable?**
>
> Our $m=k$ experiment (Section 3.1) directly demonstrates this: with fully orthogonal features, the linear classifier exhibits **zero adversarial vulnerability**. Orthogonal features eliminate interference, so perturbations changing predictions must genuinely change ground truth.
>
> However, Section 5 establishes that orthogonality is **not sufficient** for robustness in general. We demonstrate "algorithmic brittleness" as a distinct mechanism where orthogonal representations remain vulnerable when downstream computations depend sensitively on precise feature values.
>
> ### **References**
>
> [1] *Scaling and evaluating sparse autoencoders*, Gao et al., ICLR, 2025.

---

### Official Review · Reviewer_wywU · 2025-11-05

**Soundness:** 2
**Presentation:** 3
**Contribution:** 2
**Rating:** 4
**Confidence:** 3

**Summary:**

The paper argues that many adversarial examples can be explained via feature superposition: when networks pack more features than dimensions, non-orthogonal directions create interference that adversarial attacks can exploit.

In toy settings, the authors show PGD perturbations closely match theoretically optimal perturbations determined by the latent geometry, and they show an optimal input direction linking decision boundaries to input-space attacks.

The paper further shows that input correlations constrain learned geometries across seeds, which in turn predicts attack transferability between models with similar arrangements.

Moving to a ViT on CIFAR-10 with an inserted bottleneck, higher superposition (smaller m) reduces robust accuracy and increases transferability, mirroring the toy results.

The paper reframes some adversarial vulnerability as a result of representational compression, and not just non-robust features.

**Strengths:**

- The paper shows the insights of feature superposition in toy and real models. The ViT+CIFAR-10 bottleneck testbed reproduces the toy-model trends (robust accuracy down, adv transfer up as m decreases)
- The paper is easy to read and organized clearly with setups descriptions and findings.
- The source code is attached in the supplementary for reproducibility.

**Weaknesses:**

- It would be great to show more on the realism of the ViT setting, the paper shows forcing superposition via a post-hoc bottleneck which is informative but feels somewhat engineered; it would be stronger to show the same geometry/transfer phenomena in an unmodified backbone (or at higher resolution / vision datasets beyond CIFAR-10).
- It would help the readers if some more discussion/experiments on diverse architectures are presented on real world high res computer vision datasets.  Transfer measurements are mainly across seeds of the same family; it’d be valuable to test cross-architecture transfer to stress the “shared geometry” claim.
- Apart from the certified-training in the arithmetic task baseline, there’s limited comparison against adversarial trained model or modern certified/empirical defenses on the vision benchmark.

**Questions:**

- In figure 4, I was wondering how is the transfer attack percentage calculated, is it on the subset of test set where the model is correct, as the clean accuracy would decrease with decreasing m ?
- In figure 4, What would happen if bottleneck m is more than 10 for cifar10, does it remain the same as 10?

---

> ### Author Response · Authors · 2025-11-29
> **Comment by Authors (Part 1)**
>
> We thank the reviewer for their constructive feedback and thoughtful questions. We respond to each of their points below.
>
> ### **W1. Increasing the realism of the ViT setting with an unmodified backbone**
>
> We focused on the encoder/decoder bottleneck setup because it is an illustrative way to investigate the contribution of superposition to adversarial vulnerability and minimise confounding factors.
>
> We agree extending to a more representative model would further demonstrate generalisability. We therefore present an additional experiment extending our argmax task to a standard transformer architecture. The advantages of this are:
>
> - **Realistic mechanisms**: The model includes standard ViT components (CLS token, attention, residual stream, etc.)
> - **Maintains ground truth features**: We retain the necessary simplification of analysing known 'class features'. This core assumption used throughout the paper allows us to know what the network represents, and connect input changes to latent changes.
> - **Mechanistic clarity**: Avoids the need for feature extraction techniques (e.g., SAEs) in hidden layers required for natural images, which introduce reconstruction errors and confounding factors that obscure our argument.
>
> **Setup.** We use a transformer with $K$ classes distributed across $N$ tokens. The inputs are compressed through a residual stream of dimension $m$ to force the $K$ class representations into superposition (this mirrors realistic settings where $K>m$, e.g., ImageNet's 1000 classes vs CLIP ViT-B's 512-dimensional residual stream). Each sample activates only a sparse subset of classes (probability $p$), with magnitudes uniformly sampled from [-1,1], and randomly distributed across tokens such that no single token contains sufficient information to identify the winning class. The model must therefore aggregate information across all tokens to determine which class has the highest total magnitude: $y = \arg⁡max_{⁡c} \Sigma_{t} x_{t,c}$, where $t$
> indexes tokens and $c$ indexes classes..
>
> Consistent with our toy model findings (Table 1 in paper), we observe that adversarial examples systematically exploit superposition. We measure this as the alignment between PGD-discovered attacks and optimal perturbations derived from the superposition arrangement. Attacks target the interference patterns in the embedding layer, exploiting the non-orthogonal arrangement of class representations to flip predictions. We use this as further evidence that our findings are not artifacts of the simple encoder/decoder structure but reflect properties of how superposition contributes to adversarial vulnerability.
>
> | k | m | Cosine Sim. (PGD vs Theory) | Cosine Sim. (Random Baseline) |
> |:---:|:---:|:----------------------------:|:-------------------------------:|
> | 24 | 12 | 0.77 | 0.0 |
>
> Table 1: Alignment between PGD attacks and optimal perturbation across tokens

---

> > ### Author Response · Authors · 2025-11-29
> > **Comment by Authors (Part 2)**
> >
> > ### **W2. High resolution datasets, architectures, and transfer**
> >
> > Our contribution lies in revealing the mechanism by which superposition creates adversarial vulnerability, rather than comprehensive benchmarking.  Our main purpose is to obtain insights into interesting phenomena in modern deep learning. Hence, we construct a ‘toy model’, in which we can conduct controlled experimentation, and reveal hidden mechanisms behind the interaction of adversarial robustness with superposition.
> >
> > Regarding higher resolution datasets, we believe expanding to larger class counts does not work towards this aim, jeopardising a clean analysis and reducing intuition without strengthening our core claims. While our ViT and ResNet experiments bridge the gap between toy and real models, these come at a cost of interpretation. Extending this to hidden layers requires feature extraction (e.g., SAEs); as detailed in our preliminary analysis in Appendix F, the resulting reconstruction errors introduce confounding factors that would obscure our main argument that superposition is sufficient for vulnerability.
> >
> > Regarding diverse architectures, we direct the reviewer to Appendix D.5, where we validate our ViT findings using a ResNet-18, observing similar trends. Finally, regarding cross-architecture transfer, we focus on cross-seed transfer to isolate how data correlations dictate feature arrangement, avoiding the confounding variables of architectural differences.
> >
> > We also note that ICLR is not a vision venue and our analysis spans a range of modalities from arithmetics to vision.
> >
> > ### **W3. On the comparison with adversarial training and defences**
> >
> > We emphasise that the primary contribution of this paper is to provide a mechanistic understanding of how superposition is sufficient to create adversarial vulnerability, rather than to benchmark against or propose new defences.
> >
> > Our experiments with certified training in the modular arithmetic task serve to demonstrate that defences can increase the required perturbation budget without necessarily altering the underlying attack mechanism. This reinforces our broader argument: understanding the specific mechanism of vulnerability is a prerequisite for designing principled, informed defences, rather than relying solely on norm-based metrics.
> >
> > We acknowledge the reviewer’s interest in comparisons with adversarial training. Within our framework, we hypothesise that adversarial training functions by reducing feature interference—essentially reshaping the superposition geometry—which could connect to the accuracy-robustness trade-off observed in the literature. This is a promising direction for research that we note for future investigation.
> >
> > ### **Q1. How is transfer attack percentage calculated?**
> >
> > We evaluate this on the subset of samples that are correctly classified by both the source and target models, and for which the attack was successful on the source model. The reported percentage reflects the fraction of these samples where the target model's prediction on the adversarial input differs from the true label.
> >
> > ### **Q2. What happens if bottleneck m > 10 for CIFAR-10?**
> >
> > Regarding the scenario where $m>10$, we postulate that the perturbation budget required to attack this layer of the model would increase and the underlying interference patterns would persist. As we discuss in Appendix D.4, the arrangement of latent features is likely driven by correlations in the input representations (in this case the linear inseparability between classes like 'cat' and 'dog'). Increasing the bottleneck dimension does not remove this; therefore, we expect the model to maintain a similar feature arrangement and class-specific vulnerability profile.

---

### Author Response · Authors · 2025-11-29

We thank all reviewers for their thoughtful feedback and for noting that the paper provides novel insights and is easy to read. However, we acknowledge the critical responses and use this general comment to situate our paper and defend our experimental choices.

**Core contribution:** We demonstrate superposed representations as a sufficient condition for adversarial vulnerability, enabling predictive rather than post-hoc understanding in this setting. We provide an explanation for which perturbations succeed, why attacks transfer between models, and class-specific vulnerability patterns.

**On the use of toy models:** The simplified setting is deliberate—it provides a "mechanistic microscope" where we can isolate superposition's effects with ground-truth feature labels and derive exact mechanisms. Section 4 (ViT/CIFAR-10) demonstrates these principles persist in more realistic settings.

**On generalisability:** We agree additional architectures, datasets, and cross-architecture transfer would strengthen claims. The preliminary SAE analysis (Appendix F) shows a path toward analysing unmodified large-scale models. We will take this feedback onboard going forward.

**Why this matters:** Superpositional compression persists in frontier models. Understanding how this efficient information encoding creates vulnerability represents an important perspective shift with implications for defense design. Section 5 demonstrates how mechanistic understanding enables informed attack construction and suggests defenses should consider geometric arrangements rather than relying solely on norm-bounded robustness.

We believe our paper opens new research directions at the intersection of mechanistic interpretability and robustness.

---

### Meta-Review · Area_Chair_VN5n · 2026-01-03

**Summary:**

Reviewer wywU’s major concern is that the experiments should be performed on the real-world datasets. Besides, there’s limited comparison against adversarially trained models or modern certified/empirical defenses on the vision benchmark. The authors provided additional experiments using the ViT backbone, but did not apply their methods to real-world datasets or compare with more SOTA methods.

Reviewer Pyj2 did not provide a quality review.

Reviewer SnJF also argues that the evaluation should only use the synthetic dataset.

Reviewer b9T9 gave a strong rejection rating. The main concern is that this paper only investigates adversarial samples in linear models (That linearity constraints hold in linear networks is obvious, and we do not need this analysis). The reviewer also finds the evaluation of the proposed theory unconvincing.

**Reviewer Concerns:**

The evaluation of ViT has been addressed by the authors. But more evaluation on real-world datasets is suggested.

**Reviewer Scores:**

I do not think the reviewers will change their score significantly.

---

### Decision · Program_Chairs · 2026-01-26

Reject